# C²-Evo: Co-Evolving Multimodal Data and Model for Self-Improving Reasoning

## Abstract

Recent advances in multimodal large language models (MLLMs) have shown impressive reasoning capabilities. However, further enhancing existing MLLMs necessitates high-quality vision-language datasets with carefully curated task complexities, which are both costly and challenging to scale. Although recent self-improving models that iteratively refine themselves offer a feasible solution, they still suffer from two core challenges: (i) most existing methods augment visual or textual data separately, resulting in discrepancies in data complexity (e.g., over-simplified diagrams paired with redundant textual descriptions); and (ii) the evolution of data and models is also separated, leading to scenarios where models are exposed to tasks with mismatched difficulty levels. To address these issues, we propose C²-Evo, an automatic, closed-loop self-improving framework that jointly evolves both training data and model capabilities. Specifically, given a base dataset and a base model, C²-Evo enhances them by a cross-modal data evolution loop and a data-model evolution loop. The former loop expands the base dataset by generating complex multimodal problems that combine structured textual sub-problems with iteratively specified geometric diagrams or mathematical functions, while the latter loop adaptively selects the generated problems based on the performance of the base model, to conduct supervised fine-tuning and reinforcement learning alternately. Consequently, our method continuously refines its model and training data, and consistently obtains considerable performance gains across multiple mathematical reasoning benchmarks. Our code, models, and datasets will be released upon acceptance.

## 1 Introduction

Recent advancements in large language models (LLMs) have achieved remarkable progress in solving problems, including mathematics (Cobbe et al., 2021; Hendrycks et al., 2021), coding (Chen et al., 2021; Gu et al., 2024), etc. These capabilities are enabled by advanced strategies, including chain-of-thought prompting (Wei et al., 2022), tool-augmented reasoning (Feng et al., 2025; Hu et al., 2024; Li et al., 2025), *etc*. In particular, OpenAI o1 (OpenAI, 2024) and Deepseek-R1 (Guo et al., 2025a) have shown that reinforcement learning plays a critical role in aligning model outputs with desired behaviors by using structured reward signals derived from correctness, consistency, or human preference. This mechanism has proven especially effective in eliciting nuanced self-verification and self-correction behavior in LLMs, thereby reinforcing the reliability and depth of their reasoning chains, particularly in mathematical and logical domains.

Despite these advances, achieving such strong reasoning performance remains heavily reliant on large-scale, high-quality, and complexity-aligned datasets. As task complexity increases, collecting suitable training data becomes significantly more costly and difficult, presenting a major bottleneck to further progress. This challenge has sparked growing interest in *self-improving* paradigms, where models iteratively enhance their capabilities by generating new synthetic data and refining reasoning traces.

Recent studies have shown that reasoning abilities can be substantially improved through carefully curated and progressively challenging data. For example, OpenVLThinker (Deng et al., 2025b) adapts the self-improvement paradigm to the vision-language domain by iteratively alternating between

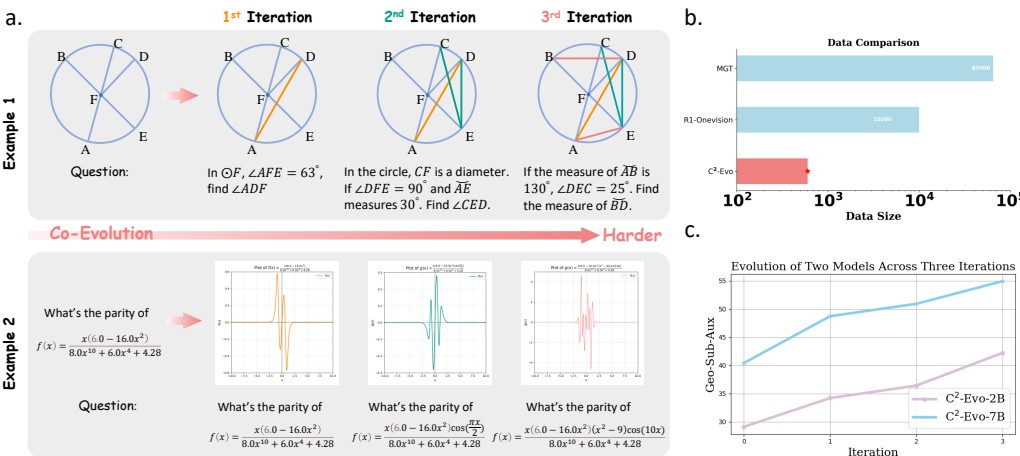

Figure 1: **a:** Co-evolution of visual-textual pairs with escalating task difficulty over three iterations. **b:** Data usage comparison among different methods. Note that the x-axis (gate count) is on a log-scale. **c:** Performance of 2B and 7B models across three iterations.

supervised fine-tuning (SFT) and reinforcement learning (GRPO), distilling R1-style reasoning traces from text-based models into multimodal contexts.

However, despite these promising developments, existing approaches face two key limitations: (1) **Mismatched evolution of visual complexity and textual reasoning difficulty**. Prior methods often suffer from decoupled difficulty scaling, where the evolution of visual and textual complexities is asynchronous. Some approaches prioritize visual complexity but fail to match it with deep reasoning tasks, while others enhance textual semantics but are constrained by static visual sources. This disconnect restricts the model's ability to learn integrated cross-modal reasoning strategies. (2) **The discrepancy between model capability and task difficulty**. As models improve over the course of training, their ability to tackle more complex tasks naturally increases. However, current approaches rely on static or manually defined difficulty schedules, which do not adapt to the model's evolving capability. This misalignment can lead to inefficient training, either under-challenging the model or overwhelming it with excessively difficult data.

Table 1: Comparison of evolutionary strategies.

| Model | Visual Evolve | Text Evolve | Capability-Aligned |
|---|---|---|---|
| MindGYM (Xu et al., 2025) | ✗ | ✓ | - |
| R-CoT (Deng et al., 2024) | ✓ | ✗ | - |
| MAVIS (Zhang et al., 2024b) | ✓ | ✗ | - |
| OpenVLThinker (Deng et al., 2025b) | - | - | ✗ |
| C$^2$-Evo(Ours) | ✓ | ✓ | ✓ |

To address these challenges, we propose a fully automated, adaptive multimodal learning framework (C$^2$-Evo) that jointly evolves both the model and its training data in a closed-loop fashion, with a particular focus on diagram-based mathematical reasoning tasks. Table 1 summarizes the differences between our framework and state-of-the-art methods: Unlike existing methods that rely on static data, our method dynamically adjusts task complexity based on real-time assessments of model performance, ensuring a tighter coupling between model capability and data difficulty throughout the learning process. Specifically, to tackle **the challenge (1)**, we incorporate the process into a cross-modal data evolution loop, where complex visual elements (*e.g.*, geometric diagrams or function plots) are jointly synthesized with semantically aligned textual problems. This is achieved by leveraging an external reasoning engine to generate and render visual augmentations, followed by the automatic construction of multi-step reasoning questions grounded in the generated visuals. The resulting multimodal samples are filtered and validated to ensure internal consistency and cross-modal coherence, yielding a dataset in which visual and textual complexities are jointly calibrated. For **the challenge (2)**, we adopt a data-model evolution loop. This loop utilizes data generated from the cross-modal data evolution loop and applies it to iteratively fine-tune the model using SFT and GRPO. SFT maintains output structure and coherence, while GRPO improves generalization through rule-based optimization. We introduce a simple error-based filtering method that evaluates sample difficulty via prediction variance over 32 generations. By selecting samples with an general error rate (*e.g.*, 0.3), which are prone to error yet still within the model's grasp and pose a meaningful

challenge, the framework ensures that task difficulty remains aligned with model capability, achieving continuous improvement over iterations (*cf.* , Figure 1 (c)).

Finally, we investigate the impact of different data strategies and iterative training regimes on model performance. Our findings offer insights into the design of effective self-improving frameworks for improving complex diagram-based mathematical reasoning and guiding the progressive evolution of vision-language models.

Our contributions are summarized as follows:

• We propose a closed-loop self-improving framework, named $C^2$-Evo, that jointly evolves training data and model capabilities.

• The proposed framework utilizes two co-evolution loops to improve the compatibility of cross-modal complexity and that between task difficulty and model capability.

• Extensive experiments (*e.g.*, Geo-Sub, MathVista, MathVerse) demonstrate the effectiveness of different data strategies and iterative training regimes, revealing their impact on self-improving frameworks.

## 2 RELATED WORK

**Self-Improvement.** Self-improvement (Fernando et al., 2023; Bhattarai et al., 2024; Rosser & Foerster, 2025) is a paradigm in which models generate and train on synthetic data generated from the same or other models. While self-improvement has been widely studied in the NLP domain, several works (Zelikman et al., 2022; Gulcehre et al., 2023; Singh et al., 2023; Lupidi et al., 2024; Liang et al., 2024; Costello et al., 2025) have explored the approach of first generating high-quality data and subsequently fine-tuning models on this data to achieve continuous performance improvement. For example, STaR (Zelikman et al., 2022) introduces a bootstrapping mechanism that enhances LLM reasoning capabilities by iteratively generating and filtering "chain-of-thought" rationales, then fine-tuning the model on correct rationales to progressively improve performance. ReST (Gulcehre et al., 2023) integrates self-generated data with offline RL, alternating between a "Grow" phase that expands the dataset by generating multiple outputs per input, and an "Improve" phase that ranks and filters these outputs using a reward model based on human preferences. Other works (Lupidi et al., 2024) follow a similar approach of generating synthetic data, filtering out low-quality samples, and fine-tuning models on the filtered high-quality data. Recent efforts (Deng et al., 2024; Zhang et al., 2024b; Trinh et al., 2024; Luo et al., 2025; Guo et al., 2025b; Fang et al., 2024) (*e.g.*, MMEvol) have introduced synthetic data generation pipelines that leverage a visual data engine to produce visual images along with corresponding reasoning tasks. However, these methods primarily focus on expanding the training distribution. Our approach shifts the emphasis from passive data enrichment to self-improvement with a co-evolution mechanism. A closely related work is OpenVLThinker (Deng et al., 2025b), which introduces an iterative training paradigm in which models are exposed to progressively more complex tasks. However, their approach relies on manually defined task difficulty, which does not adapt to the evolving capabilities of the model.[1]

## 3 METHOD

In this section, we present the details of the proposed $C^2$-Evo framework for self-improving reasoning. The key idea behind $C^2$-Evo is the joint evolution of multimodal data and models, thereby mitigating discrepancies not only between visual and textual modalities but also between task difficulty and model capabilities. We begin by formulating our task in Sec. 3.1, and then introduce the two core components of $C^2$-Evo, namely the multimodal data co-evolution loop (Sec. 3.2), and the data and model co-evolution loop (Sec. 3.3), as shown in Figure 2.

### 3.1 TASK FORMULATION

In this paper, we are given a pre-trained multimodal large language model (MLLM) denoted as $\pi_\theta$ and a dataset $\mathcal{D}$ consisting of image-question-answer triplets, namely, $\mathcal{D} = \{D_n = (I^n, Q^n, G^n) | n =$

---

[1]Additional Related Work is provided in the **APPENDIX A.1**.

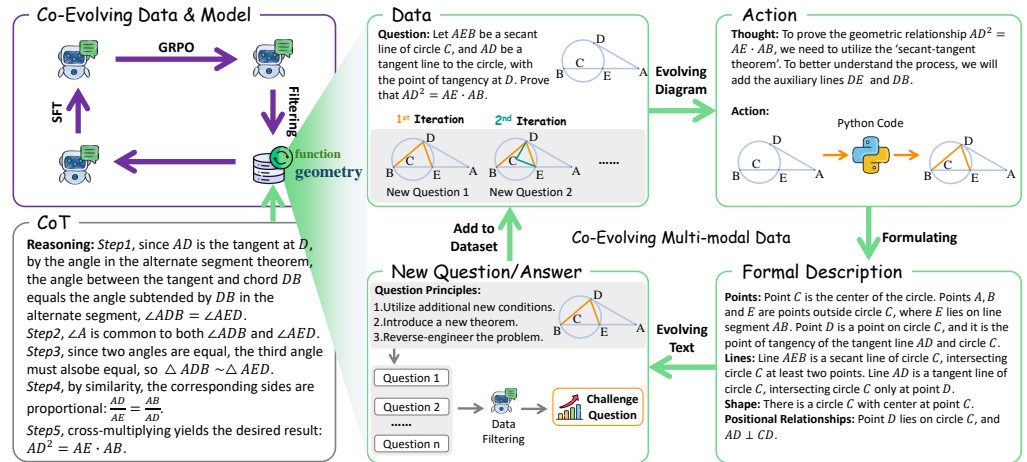

Figure 2: **The proposed C²-Evo framework**. Starting from a base dataset and a base model, a co-evolving multimodal data loop iteratively synthesizes complex paired visual and textual samples. These samples are subsequently filtered and utilized in the co-evolving data & model loop to iteratively improve the reasoning performance of the model.

$1, ..., |\mathcal{D}|\}$, where $I^n$, $Q^n$, and $G^n$ denote an image, questions related to $I^n$, and the corresponding answers of $Q^n$, respectively. Our goal is to improve the complex reasoning capability of $\pi_\theta$ by optimizing its parameters $\theta$ on $\mathcal{D}$ within $T$ iterations. Following recent studies (Deng et al., 2024; Zhang et al., 2024b; Hu et al., 2024), we assume that the necessary external tools and models are available, such as an oracle model that can generate and execute Python code to edit images (Hu et al., 2024).

Specifically, let $t = 1, ..., T$ denote the iteration step. At the beginning of the $t$-th iteration, we first conduct the multimodal data co-evolution loop to augment $\mathcal{D}_t$. This is achieved by prompting an MLLM (Hurst et al., 2024) as the oracle model for solving each question in $\mathcal{D}_t$, generating step-by-step reasoning trajectories including necessary image augmentations like drawing auxiliary lines. For an arbitrary triplet $D_t^n \in \mathcal{D}_t$, the MLLM produces an action sequence $Ac_t^n$, corresponding Python code segment $C_t^n$, and a natural language reasoning trace $R_t^n$. The generated code $C_t^n$ is then executed using an external image editing tool (Jupyter) to yield a more complex image $I_{t+1}^n$ that includes auxiliary augmentations. To ensure that the questions align with $I_{t+1}^n$ with increased complexity, more challenging questions $Q_{t+1}^n$ and answers $G_{t+1}^n$ regarding $I_{t+1}^n$ are generated and curated, consequently a new multimodal triplet $D_{t+1}^n = (I_{t+1}^n, Q_{t+1}^n, G_{t+1}^n)$ is obtained. We assess the difficulty of $D_{t+1}^n$ and if it is sufficiently challenging, we incorporate it into $\mathcal{D}_t$. Afterwards, we utilize the updated dataset $\mathcal{D}_{t+1}$ to train the base model $\pi_\theta$ through a combination of SFT and GRPO, where SFT establishes the initial reasoning structure and GRPO improves its generalization capability.

## 3.2 CO-EVOLVING MULTIMODAL DATA

The co-evolving multimodal data loop is introduced to mitigate the complexity discrepancy between visual and textual data, which can be further divided into an action and reasoning generation process and a challenging question generation process.

**Action and Reasoning Generation.** Inspired by SKETCHPAD (Hu et al., 2024), we introduce a two-stage framework designed to enable the precise construction of auxiliary lines in geometric diagrams. We first extract the coordinates of key points using Optical Character Recognition (OCR) and other vision-based techniques. This coordinate-level representation ensures robustness and consistency in subsequent diagram transformations, particularly under increased geometric complexity. Leveraging this representation and image $I_t$ (we omit the superscript $n$ for conciseness), we prompt[2] GPT-4o to determine whether auxiliary image augmentations are needed to facilitate problem solving. This

---

[2]All detailed prompts are provided in the **APPENDIX A.6.3**.

---

**Algorithm 1** C²-Evo Algorithm

---

**Input:** Seed Dataset $\mathcal{D}$, Base Model $\pi_\theta$, Principles $P$, Number of Iterations $T$
**Output:** Improved Model $\pi_{\theta,\mathrm{RL}}^{T+1}$, Evolved Datasets $D_{T+1}$

---

1: Initialize $\pi_\theta^1 = \pi_\theta$
2: **for** $t = 1$ to $T$ **do**
3:     ▷ **Co-Evolving Multimodal Data**                             ▷ see §3.2
4:     **for** $n = 1$ to $|\mathcal{D}_t|$ **do**
5:         $Ac_t^n, C_t^n, R_t^n \leftarrow$ generate action and reasoning conditioned on $D_t$
6:         $I_{t+1} \leftarrow$ apply auxiliary augmentations by executing $(Ac_t^n, C_t)$
7:         $Q_{t+1}, A_{t+1} \leftarrow$ generate challenging questions and answers conditioned on $I_{t+1}, P, Q_t$
8:         $D_{t+1} \leftarrow$ add filtered tuple $(I_{t+1}, Q_{t+1}, A_{t+1})$
9:     **end for**
10:   ▷ **Co-Evolving Data and Model**                               ▷ see §3.3
11:   $D_t^{\mathrm{train}} \leftarrow$ select data from $D_{t+1}$ using error rate evaluated with $\pi_{\theta,\mathrm{RL}}^t$
12:   $\pi_{\theta,\mathrm{SFT}}^{t+1} \leftarrow$ update $\pi_\theta^t$ with SFT on $D_t^{\mathrm{train}}$          ▷ using Equation 1
13:   $\pi_{\theta,\mathrm{RL}}^{t+1} \leftarrow$ update $\pi_{\theta,\mathrm{SFT}}^{t+1}$ with GRPO on $D_t^{\mathrm{train}}$     ▷ using Equation 2
14: **end for**

---

generates a decision, denoted as *Thought*, indicating whether auxiliary augmentations are necessary and specifying their types (*e.g.*, parallel lines, perpendicular lines, connecting lines and function graph). Subsequently, it produces the updated image $I_{t+1}$ with newly added auxiliary augmentations, as illustrated in Figure 2. In addition to $I_{t+1}$, the pair $(I_t, Q_t)$ is also fed to GPT-4o to simultaneously generate the full reasoning trace $R_t$, which is used in the subsequent generation of challenging questions. For mathematical functions, we prompt GPT-4o to decide whether plotting the graph would aid in solving the given problem. The rest of the process follows the same procedure as before.

**Challenging Question Generation.** This step aims to match the level of problem difficulty with the degree of image complexity. For geometric diagrams, we utilize the GPT-4o to generate a formatted description $F_t$ conditioned on the generated images $I_{t+1}$. To promote diversity in the generated sub-problems, we define a set of guiding principles $P$:

1) *Math Constraints*: We extract mathematical constraints (*e.g.*, perpendicularity, equality, and sub-images) from the formal description $F_t$.

2) *New Theorems and Concepts*: We incorporate relevant mathematical theorems and conceptual principles (*e.g.*, the properties of a right triangle imply the Pythagorean theorem, symmetry, parity) that are closely related to the formal description $F_t$.

3) *Backward Reasoning*: We also include the concept (*e.g.*, the Tangent-Secant theorem) of sub-steps in the inference trace (*i.e.*, $R_t$).

Using these principles, we prompt GPT-4o with the tuples $(F_t, R_t, P)$ (*e.g.*, $F_t$ is applied exclusively to geometric diagrams) to generate a diverse set of sub-problems $(q_1, q_2, \ldots, q_m$, where $m$ typically ranges from 4 to 10, depending on the complexity of the image). To mitigate the inherent limitations of formal language in capturing visual content, we incorporate the role-playing strategy from R1-Onevision (Yang et al., 2025a). By analyzing the differences and commonalities among sub-problems $(q_1, q_2, \ldots, q_m)$, we align them with corresponding sub-image elements to compose challenging geometric reasoning

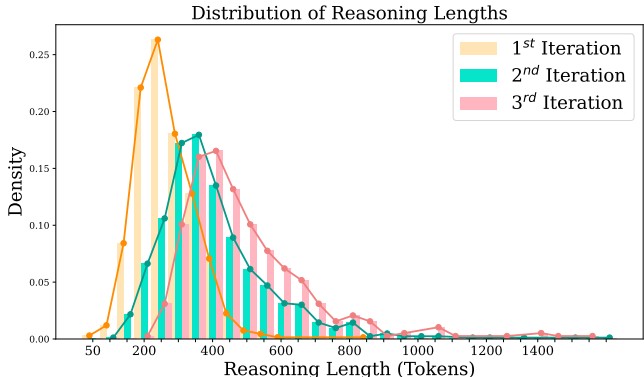

Figure 3: **Evolution of reasoning length** (tokens) over three iterations.

questions. Specifically, by combining the sub-problems $(q_1, q_2, \ldots, q_m)$ with their associated sub-image descriptions $F_t$, we prompt GPT-4o to compose challenging questions. For example, given two sub-problems $q_1$ and $q_2$ that address different parts of an image, we combine them into a more challenging question $Q_{t+1}$ by leveraging their thematic and contextual connections.

To obtain corresponding answers, we feed each generated question $Q_{t+1}$ together with its formal description $F_t$ into GPT-4o three times. For each input, the model produces a step-by-step reasoning trace $R_{t+1}$ and the final answer $A_{t+1}$. We then retain only those samples with three answers $A_{t+1,1}, A_{t+1,2}, A_{t+1,3}$ are equal. We further prompt model to filter out any questions that are inconsistent with the original image constraints, ensuring alignment between the question and the visual content. As illustrated in Figure 3, the increasing length of the reasoning trajectories serves as a reliable indicator of problem difficulty, aligning with our data evolution design.

### 3.3 Co-Evolving Data and Model

**SFT to Follow Reasoning Structure** This stage aims to unify the model's reasoning format with the structure needed during the later reinforcement learning (GRPO) phase, ensuring compatibility through a standardized template (*i.e.*, <think></think><answer></answer>). The model is trained on the previously constructed triplets $(I_{t+1}, Q_t, R_t)$, learning to generate the reasoning trace and final answer in a structured format. The corresponding training objective is formulated as follows:

$$\mathcal{L}_{\text{SFT}} = -\mathbb{E}_{(I,Q,R) \sim D}[\log(\pi_\theta(R|I,Q)]. \tag{1}$$

**Group Relative Policy Optimization** Based on the above obtained $\pi_{\theta,\text{SFT}}$ model, we employ two reward rules in conjunction with the GRPO algorithm to further refine the policy model. These rewards are designed as follows:

1) Accuracy Reward: This reward evaluates the correctness of the final answer $A$ by processing the final answer via regular expressions and verifying it against the ground truth $G$.

2) Format Reward: To ensure consistent and well-structured reasoning trajectories, we define a reward based on the model's adherence to the predefined format <think>$\cdots$</think>.

Additionally, we enforce sequential consistency within the reasoning trace by requiring that each step be explicitly arranged in the correct order. Responses that violate this ordering are penalized during training. The training objective with GRPO is defined as,

$$\mathcal{L}_{\text{RL}} = \mathbb{E}\left[\min\left(r_t(\theta)A_t, \text{clip}(r_t(\theta), 1 - \varepsilon, 1 + \varepsilon)A_t\right) - \beta D_{\text{KL}}[\pi_\theta \| \pi_{\text{old}}]\right]. \tag{2}$$

**Iterative Refinements and Filtering**

In the $t$-th iteration, we perform SFT followed by GRPO training using all post-filtered data, resulting in an updated model $\pi_{\theta,RL}^{t+1}$ and a new dataset $\mathcal{D}_{t+1}$ for the next round.

To align the difficulty of $\mathcal{D}$ with the current capability of $\pi_\theta$, we forward each sample through the model 32 times and compute the error rate as the proportion of times the model generated a wrong answer. As illustrated in Figure 4, we evaluate the difficulty of the data for each model obtained from GRPO training. We then retain only the samples with an error rate greater than $0.3$ to form the training set for the next iteration, ensuring that the data complexity remains aligned with model capability. More assessments are provided in **APPENDIX A.6.2**.

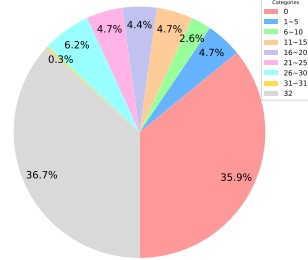

Figure 4: Assessment of third iteration training data difficulty using second iteration model $\pi_{\theta,\text{RL}}^2$.

## 4 Experiment

### 4.1 Benchmark

**Dataset**. We use the Geometry3k (Lu et al., 2021) dataset as the base dataset for multimodal data evolution. Geometry3k is a mathematical benchmark dataset that comprises 3,002 geometry problems, divided into 2,101 training examples, 300 for validation, and 601 for testing. Each problem

Table 2: Main experimental results. For MathVista benchmark, we have specifically compared all models on three sub-tasks that are highly related to mathematical reasoning: geometry reasoning (GEO), algebraic reasoning (ARI) and geometry problem solving (GPS). The result ($\dagger$) is collected from original papers, R1-VL(Zhang et al., 2025) and R1-Onevison (Yang et al., 2025a). The remaining results are reproduced under the same experimental setting.

| Methods | Data Amount | Geo-Sub | Geo-Sub-Aux | MathVista | | | |
| --- | --- | --- | --- | --- | --- | --- | --- |
| | | | | GEO | ARI | GPS | ALL |
| *Closed-Source Model* | | | | | | | |
| GPT-4o (Hurst et al., 2024) | - | - | - | - | - | - | 63.8$\dagger$ |
| *Reasoning Model* | | | | - | | | |
| LLamaV-o1-11B(Thawakar et al., 2025) | >100k | - | - | - | - | - | 54.4$\dagger$ |
| Insight-V-8B(Dong et al., 2024) | 200K | - | - | - | - | - | 49.8$\dagger$ |
| MGT-PerceReason (Peng et al., 2025b) | 65k | - | - | - | - | - | 63.2$\dagger$ |
| R1-Onevision-7B (Yang et al., 2025a) | 10k | - | - | - | - | - | 64.1$\dagger$ |
| R1-VL-7B (Zhang et al., 2025) | 10k (SFT 120k) | - | - | - | - | - | 63.6$\dagger$ |
| Qwen2-VL-2B (Wang et al., 2024) | - | 28.0 | 29.1 | - | - | - | 43.0$\dagger$ |
| C$^2$-Evo-1$^{st}$ | 0.6k | 32.0 | 34.2 | 38.0 | 40.0 | 38.0 | 49.1 |
| C$^2$-Evo-2$^{nd}$ | 0.6k | 35.6 | 36.4 | 38.0 | 38.0 | 38.0 | 49.3 |
| C$^2$-Evo-3$^{rd}$ | 0.4k | 38.2 | 42.2 | 40.0 | 40.0 | 38.0 | 50.2 |
| Qwen2-VL-7B (Wang et al., 2024) | - | 40.4 | 40.4 | 50.0 | 50.0 | 49.0 | 60.0 |
| C$^2$-Evo-1$^{st}$ | 0.6k | 45.5 | 46.9 | 55.0 | 57.0 | 54.0 | 62.1 |
| C$^2$-Evo-2$^{nd}$ | 0.6k | 46.9 | 48.0 | 56.0 | 58.0 | 56.0 | 62.4 |
| C$^2$-Evo-3$^{rd}$ | 0.4k | 50.9 | 52.4 | 59.0 | 59.0 | 60.0 | 63.2 |

is accompanied by a corresponding geometric diagram, a natural language description, and formal language annotations.

**Implementation Details**. In our experiments, we adopt two state-of-the-art open-source MLLMs, *i.e.*, Qwen2-VL-2B (Wang et al., 2024) and Qwen2-VL-7B (Wang et al., 2024). For the policy warm-up phase, we employ the LLaMA-Factory. framework with a batch size of 128 and a learning rate of $1e-5$. For the GRPO phase, we use the VLM-R1. framework. In the first iteration, we perform 32 rollouts per question, reducing to 8 in subsequent iterations. The temperature is set to the default value of 0.9, and the KL divergence coefficient $\beta$ in Equation 2 is set to 0. All experiments are conducted on 32 NVIDIA V100-32GB GPUs.

**Evaluation Settings**. We evaluate C$^2$-Evo on several multimodal reasoning benchmarks, including Geo-Sub from Geometry3k-test (Lu et al., 2021), MathVista (Lu et al., 2023), and MathVerse (Zhang et al., 2024a). To provide a more comprehensive evaluation of model performance, we construct a specialized test subset by sampling images from Geometry3K-test that require the use of auxiliary lines to solve. This results in a focused benchmark containing 274 images. When evaluated on the original images, we denote the setting as Geo-Sub. Using the corresponding images with added auxiliary lines, we refer to the setting as Geo-Sub-Aux. Further details can be found in the **APPENDIX A.2**.

## 4.2 MAIN RESULTS

In the main paper, we only present a portion of the experimental results; additional experiments are provided in the appendix (*e.g.*, **APPENDIX A.3, A.4, A.6**).

As shown in Table 2, C$^2$-Evo demonstrates a significant improvement over Qwen2-VL-2B and Qwen2-VL-7B across three bench-

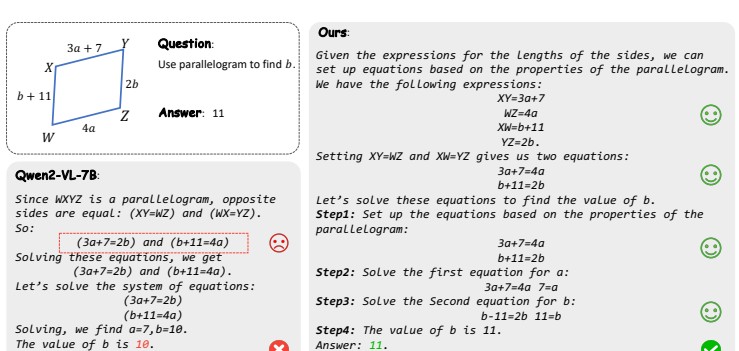

Figure 5: Qualitative comparison of geometric reasoning.

marks through three iterations, with geometric reasoning accuracy increasing notably from $29.1$ to $42.2$ and $40.4$ to $52.4$, respectively. The progressive improvement observed across three iterations highlights the effectiveness of our self-improving strategy in enhancing reasoning capabilities. Furthermore, compared to prior reasoning models, our approach achieves superior performance while leveraging less than $1\%$ of the training data, with performance approaching that of GPT-4o.

Figure 5 illustrates the model's behavior on a geometry problem involving a parallelogram. The response generated by Qwen2-VL-7B is relatively short but lacks a thorough reasoning process. It contains inaccuracies in both formula application and computational steps, leading to an incorrect final answer. In contrast, our $C^2$-Evo produces a well-structured and logically coherent solution. It begins

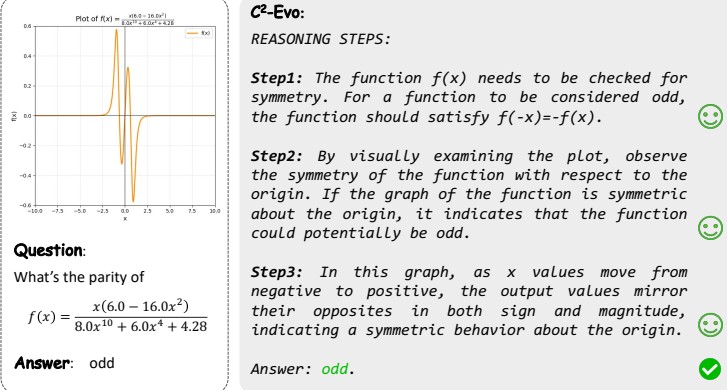

Figure 6: Visualization of the Mathematical Function.

with a thorough analysis of the given conditions, followed by a step-by-step deductive process that ensures correctness at each stage, ultimately yielding the accurate final result. Figure 6 presents the results of $C^2$-Evo evaluation on mathematical function reasoning tasks. The findings demonstrate that our model exhibits robust and accurate reasoning capabilities in handling mathematical problems, indicating its proficiency in comprehensively understanding and correctly analyzing mathematical functions.

Table 3: Comparison of alternating vs. consecutive GRPO training schedules with progressive model updates.

| Iteration | Model | SFT | GRPO | Geo-Sub-Aux |
|---|---|---|---|---|
| Second | $\pi_\theta^1$ | ✔ | ✔ | 40.7 |
| | $\pi_\theta^1$ | - | ✔ | **48.0** |
| Third | $\pi_\theta^2$ | ✔ | ✔ | 42.9 |
| | $\pi_\theta^2$ | - | ✔ | **52.4** |

Table 4: Comparison of alternating vs. consecutive GRPO training schedules from the initial model.

| Iteration | Model | SFT | GRPO | Geo-Sub-Aux |
|---|---|---|---|---|
| Second | $\pi_\theta^0$ | ✔ | ✔ | **48.4** |
| | $\pi_\theta^0$ | - | ✔ | 44.0 |
| Third | $\pi_\theta^0$ | ✔ | ✔ | **46.7** |
| | $\pi_\theta^0$ | - | ✔ | 44.4 |

**The influence of different iteration strategies.** Table 3 presents the results of different iteration strategies. Fine-tuning $\pi_\theta^1$ with additional SFT in the second iteration leads to performance degradation. In contrast, further refining the model through reinforcement learning yields improved results. As shown in Table 4, when the second iteration model is trained via SFT and RL starting from the initial model $\pi_\theta$, the same performance degradation is not observed. However, the resulting model underperforms compared to when the training is warm-started from $\pi_\theta^1$. Similar trends are observed in the third iteration.

**The Influence of error-rate.** As described in Section 3.3, we conduct an extensive parameter analysis over varying values of error-rate, which governs the complexity of data within each iteration. Table 5 shows the performance of models trained solely on data from the current iteration $D_t$, compared to those that also incorporate historical data $D_{1,\ldots,t-1}$. Utilizing the complete dataset does not necessarily optimize the model's reasoning capabilities. Specifically, as the complexity of the training data increases, relying exclusively on error-prone samples results in a

Table 5: Comparison of error-rates based on the second iteration.

| Models | Data Size ($D_t/D_{1\sim t}$) | Geo-Sub-Aux |
|---|---|---|
| Qwen2-VL-7B | - | 38.2/40.4 |
| Fullset | 0.83K/1.48K | 48.4/46.6 |
| $\pi_\theta^1$- 0.3 | 0.48K/0.64K | 49.1/**48.0** |
| $\pi_\theta^1$- 0.6 | 0.44K/0.54K | **50.9**/47.3 |
| $\pi_\theta^1$- 0.9 | 0.38K/0.44K | 48.4/47.3 |
| $\pi_\theta^1$- 1.0 | 0.34K/0.35K | 46.2/46.5 |

decline performance. Based on this analysis, error-rate$\geq 0.3$ is adopted to as the basis for our final setting. Additional generalization experiments for this selection are presented in **APPENDIX A.4.2**.

**Generalization across tasks.** We also evaluate our method on mathematical function reasoning tasks to demonstrate its broader applicability, reporting performance on FunctionQA and Function-Plot from MathVista after two evolutionary rounds. As shown in the table 6, our method also achieves strong performance on other tasks.

Table 6: The results on the mathematical function task.

| Methods | Data Amount | FunctionQA | Function-Plot |
|---|---|---|---|
| Qwen2-VL-7B | - | 58.0 | 58.0 |
| $C^2$-Evo-$1^{st}$ | 0.6k | 60.0 | 59.0 |
| $C^2$-Evo-$2^{nd}$ | 0.6k | 66.0 | 69.0 |
| $C^2$-Evo-$1^{st} + Function$ | 0.7k | 61.0 | 60.0 |
| $C^2$-Evo-$2^{nd} + Function$ | 0.7k | 72.0 | 71.0 |

**Effectiveness demonstrated on generalization metrics.** To further demonstrate the generalization capability of our method and examine potential overfitting or degradation, we evaluated our model on more comprehensive benchmarks (*e.g.*, MMMU and MME) over three iterative rounds. The results, presented in the Table 7, show that although our model is trained exclusively on geometric problems, it progressively acquires more general capabilities throughout the iterative process.

Table 7: Comparison of error-rates based on the second iteration.

| Methods | Data Amount | MMMU | MME-sum |
|---|---|---|---|
| Qwen2-VL-7B | - | 52.0 | 2320.8 |
| C2-Evo | 0.6k | 53.3 | 2335.3 |
| C2-Evo | 0.6k | 54.2 | 2336.2 |
| C2-Evo | 0.4k | 55.2 | 2337.1 |

**Compared to other methods.** Due to the authors have recently released their code and **partial** data (*e.g.*for the fine-tuning and reinforcement learning stages). We have since evaluated their model under their experimental setup (*e.g.*, besides changing the model from Qwen2.5-VL-7B to Qwen2-VL-7B.), and the results are presented in Table 8. (*e.g.*, Note that their data volume is 5k, which is significantly larger than our 0.6k. )

Table 8: Results of other self-improvement methods.

| Methods | Data Amount | Geo-Sub-Aux | MathVista |
|---|---|---|---|
| OpenVLThinkder-Medium | 5k (SFT 5k) | 37.8 | 60.7 |
| OpenVLThinkder-Hard | 5k (SFT 5k) | 37.5 | 60.3 |
| C2-Evo | 0.6k | 46.9 | 62.1 |
| C2-Evo | 0.6k | 48.0 | 62.4 |
| C2-Evo | 0.4k | 52.4 | 63.2 |

**The Influence of training data.** Table 9 presents a comparison of the 7B model trained on complexified data (complex images with complex text) versus the original data (original images with complex text). The results show that jointly complexifying both images and text leads to better performance, highlighting the importance of image complexity in training.

Table 9: Training comparison between original and complex image datasets.

| Model | Original Data | Complex Data |
|---|---|---|
| $C^2$-Evo-$1^{st}$ | 47.27 | 46.9 |
| $C^2$-Evo-$2^{nd}$ | 45.82 | 48.0 |
| $C^2$-Evo-$3^{rd}$ | 51.27 | 52.4 |

## 5 CONCLUSION

In this paper, we propose a closed-loop self-improving framework ($C^2$-Evo), a multi-dimensional evolution framework that operates through two interleaved loops: **a cross-modal data evolution loop** and **a data-model co-evolution loop**. Recent studies often suffer from the decoupling of textual and visual evolution, as well as a mismatch between model capability and task difficulty. To address these limitations, we introduce a synchronized co-evolution mechanism in which auxiliary guidance is employed to progressively increase the complexity of image data, while the resulting complex visuals are used to generate increasingly challenging diagram-based mathematical reasoning tasks. This ensures the joint evolution of both modalities. Furthermore, by leveraging the error-based filtering method, the model selects samples that align with the current model's blind spots or underdeveloped reasoning capabilities, thereby maintaining consistency between data complexity and model capability through iterative training. Extensive experiments demonstrate the effectiveness of our multi-iteration evolution training strategy. The different data curation strategies and iterative mechanisms not only improve performance but also offer promising directions for future research in self-improving methods.

## 6 STATEMENT

### 6.1 ETHICS STATEMENT

This research does not involve potentially harmful insights, methodologies, or applications, and it raises no concerns regarding conflicts of interest, sponsorship, discrimination, bias, fairness, privacy, security, legal compliance, or research integrity.

### 6.2 REPRODUCIBILITY STATEMENT

**Data.** The training datasets employed in this study are detailed in Section 4.1 and **APPENDIX A.2.2**, and additional data evolution prompt templates are provided in Section **APPENDIX A.6.3**.

**Method.** To support reproducibility, we provide a detailed description of the methodology in Section 3, and further clarify the complete procedure in Algorithm 1. The implementation will be made publicly available upon acceptance.

**Performance.** All evaluations are carried out on open benchmarks, thereby ensuring the reproducibility of our results.

**Code, Models and Datasets.** We will open-source the code, models and datasets files after acceptance.

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

# A APPENDIX

## A.1 RELATED WORK

**Multimodal Large Language Models.** Multimodal Large Language Models (MLLMs) have witnessed rapid advancements in recent years, resulting in significant breakthroughs in visual understanding and cross-modal reasoning within the field of artificial intelligence. Unlike traditional Vision-Language Models (Radford et al., 2021; Jia et al., 2021) (VLMs), which are typically trained from scratch on image-text pairs, MLLMs are generally built upon powerful pretrained text-only large language models (LLMs), and then further aligned with multimodal data, such as images and videos. Representative works such as BLIP-2 (Li et al., 2023), LLaVA (Liu et al., 2023; 2024a;b), and MiniGPT-4 (Zhu et al., 2023) have demonstrated impressive zero-shot and few-shot generalization across various tasks, including image captioning, visual question answering, and image-text reasoning. LLaVA (Liu et al., 2023), for example, pioneered the use of high-quality visual instruction data generated by GPT-4 (Achiam et al., 2023) to fine-tune MLLMs, achieving significant success in visual dialogue and reasoning. This method has generated significant interest in research focused on creating multimodal datasets for tuning vision instructions (Chen et al., 2024a; Liu et al., 2024a). BLIP-3 (Xue et al., 2024) extends the single image input of BLIP-2 to interleaved multimodal data, using large-scale, high-quality, and diverse curated data with training recipes and post-training strategies to beat other contemporary competitors on various visual understanding tasks. Recently, several open-source MLLMs have significantly narrowed the performance gap with proprietary models like OpenAI's GPT-4o (Hurst et al., 2024). Among them, InternVL2 (Chen et al., 2024b), Qwen2-VL (Bai et al., 2025), SPHINX (Lin et al., 2023), and MiniCPM-V (Yao et al., 2024) are particularly notable for their expanded application coverage, achieved through richer instruction-tuning datasets. In this study, we adopt a self-improving training paradigm to enhance the reasoning capabilities of MLLMs in complex scenarios.

**Reinforcement Learning.** Recent research has demonstrated that reinforcement learning (RL) can significantly enhance the reasoning capabilities of large language models (LLMs) such as OpenAI-o1 (OpenAI, 2024). Some approaches have introduced RL-based mechanisms to facilitate test-time scaling, achieving notable success in tasks such as mathematical reasoning and code generation. Building on this progress, DeepSeek-R1 (Guo et al., 2025a) proposed a rule-based reward strategy and adopted the Group Relative Policy Optimization (GRPO) (Shao et al., 2024) algorithm, demonstrating strong performance with only a few update steps. Motivated by the success of LLMs, recent studies (Yang et al., 2025b; Huang et al., 2025; Zhou et al., 2025; Deng et al., 2025a; Peng et al., 2025c; Meng et al., 2025; Zhang et al., 2025; Peng et al., 2025a) have begun to explore the reasoning capabilities of MLLMs. For example, R1-OneVision (Yang et al., 2025b) integrates supervised fine-tuning with RL to bridge the gap between visual perception and deep logical reasoning. Vision-R1 (Huang et al., 2025) generates cold-start initialization data and employs GRPO with hard format reward functions to enhance the emergent reasoning capabilities of MLLMs. VisualThinker-R1-Zero (Zhou et al., 2025) applies the R1 style to a base MLLM without supervised fine-tuning, surpassing traditional fine-tuning methods while exhibiting "visual aha moment" behaviors. R1-VL (Zhang et al., 2025) proposes the Step-wise Group Relative Policy Optimization (StepGRPO), an online reinforcement learning framework for improving MLLMs' reasoning ability through dense, effective step-wise rewards. MM-EUREKA (Meng et al., 2025) demonstrates that rule-based reinforcement learning can be effectively extended to multimodal reasoning, enabling emergent reasoning behaviors without supervised fine-tuning and offering superior data efficiency. Curr-ReFT (Deng et al., 2025a) adopts a multi-stage curriculum learning strategy with progressively increasing difficulty to support the self-evolution of reasoning capabilities, LMM-R1 (Peng et al., 2025c) adopts a staged approach that begins with textual reasoning and advances toward complex multimodal reasoning tasks. MM-EUREKA (Meng et al., 2025) introduces a novel data filtering strategy that simultaneously removes both unsolvable and trivial cases, along with rejection samples, retaining only high-confidence instances. The effectiveness of these methods in multimodal understanding tasks can be attributed to the presence of high-quality CoT datasets and the use of R1-style RL. Nonetheless, a significant drawback persists: the disconnect between the complexity of the training data and the difficulty of the tasks, particularly the inconsistency between the complexity of visual inputs and the challenges of textual reasoning. In this paper, we introduce a self-evolving training approach that alternates between RL and SFT. Our method dynamically modifies the multimodal training data to align with the model's reasoning abilities, ensuring that the complexities of both visual and textual elements

are consistent throughout the training process. In contrast to MM-EUREKA, our method prioritizes samples that are both meaningfully challenging and conditionally accessible, striking a balance between task difficulty and model learnability.

## A.2 EXPERIMENTAL SUPPLEMENT

### A.2.1 IMPLEMENTATION DETAILS

Table 10: Training parameters of Qwen2-VL-2B/7B model.

| Parater | Qwen2-VL-2B-SFT | Qwen2-VL-2B-RL | Qwen2-VL-7B-SFT | Qwen2-VL-7B-RL |
|---|---|---|---|---|
| Learning Rate | 1e-5 | 1e-6 | 1e-5 | 1e-6 |
| Epochs | 2 | 2 | 2 | 2 |
| Batch Size | 128 | 128 | 128 | 128 |
| Precision | fp16 | fp16 | fp16 | fp16 |
| GPU | 32 V100 | 32 V100 | 32 V100 | 32 V100 |
| Temperature | - | 0.9 | - | 0.9 |

In this section, we provide more implementation details for the $C^2$-Evo. In Table 10, we provide the training parameters of Qwen2-VL-2B/7B during the SFT and RL stages to facilitate future work. We evaluate $C^2$-Evo on three reasoning benchmarks, Geo-Sub from Geometry3K-test (Lu et al., 2021), MathVista (Lu et al., 2023), and MathVerse (Zhang et al., 2024a). For MathVista, we use the Test Mini split (*cf.* , around 1,000 samples). For MathVerse, we use the full dataset.

### A.2.2 THE DIFFERENCE BETWEEN GEO-SUB AND GEO-SUB-AUX.

We have introduced auxiliary lines into the Geo-Sub-Aux dataset to facilitate problem solving and improve model alignment during training. Originating from Geometry3k(Lu et al., 2021), the Geo-Sub dataset comprises 275 samples. The left image illustrates the structure of the Geo-Sub dataset (*cf.* , Figure 7 left), whereas the right image showcases the Geo-Sub-Aux dataset (*cf.* , Figure 7 right), highlighted by its additional auxiliary lines.

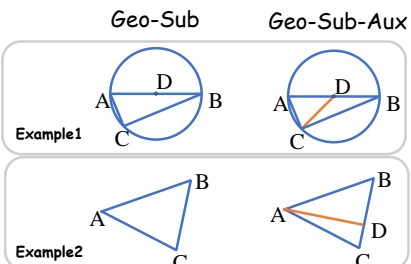

Figure 7: Data comparison between Geo-Sub and Geo-Sub-Aux.

## A.3 MORE MAIN EXPERIMENTAL RESULTS.

Table 11: Main experimental results. For MathVista benchmark, we have specifically compared all models on three sub-tasks that are highly related to mathematical reasoning: geometry reasoning (GEO), algebraic reasoning (ARI) and geometry problem solving (GPS). The result ($^\dagger$) is collected from original papers or R1-VL(Zhang et al., 2025). The remaining results are reproduced under the same experimental setting.

| Methods | Data Amount | Geo-Sub | Geo-Sub-Aux | MathVista GEO | ARI | GPS | ALL | MathVerse |
|---|---|---|---|---|---|---|---|---|
| *Closed-Source Model* | | | | | | | | |
| GPT-4o(Hurst et al., 2024) | | - | - | - | - | - | $63.8^\dagger$ | $39.4^\dagger$ |
| *Reasoning Model* | | | - | | | | | |
| LLamaV-o1-11B(Thawakar et al., 2025) | >100k | - | - | - | - | - | $54.4^\dagger$ | - |
| Insight-V-8B(Dong et al., 2024) | 200K | - | - | - | - | - | $49.8^\dagger$ | - |
| MGT-PerceReason (Peng et al., 2025b) | 65k | - | - | - | - | - | $63.2^\dagger$ | $41.6^\dagger$ |
| R1-Onevision-7B (Yang et al., 2025a) | 10k | - | - | - | - | - | $64.1^\dagger$ | $46.4^\dagger$ |
| Qwen2-VL-2B(Wang et al., 2024) | - | 28.0 | 29.1 | - | - | - | $43.0^\dagger$ | 17.3 |
| $C^2$-Evo-$1^{st}$ | 0.6k | 32.0 | 34.2 | 38.0 | 40.0 | 38.0 | 49.1 | 16.7 |
| $C^2$-Evo-$2^{nd}$ | 0.6k | 35.6 | 36.4 | 38.0 | 38.0 | 38.0 | 49.3 | 18.5 |
| $C^2$-Evo-$3^{rd}$ | 0.4k | 38.2 | 42.2 | 40.0 | 40.0 | 38.0 | 50.2 | 19.1 |
| Qwen2-VL-7B(Wang et al., 2024) | - | 40.4 | 40.4 | 50.0 | 50.0 | 49.0 | 60.0 | 30.2 |
| $C^2$-Evo-$1^{st}$ | 0.6k | 45.5 | 46.9 | 55.0 | 57.0 | 54.0 | 62.1 | 33.6 |
| $C^2$-Evo-$2^{nd}$ | 0.6k | 46.9 | 48.0 | 56.0 | 58.0 | 56.0 | 62.4 | 34.8 |
| $C^2$-Evo-$3^{rd}$ | 0.4k | 50.9 | 52.4 | 59.0 | 59.0 | 60.0 | 63.2 | 34.9 |

Table 12: Comparison of SFT vs. RL on first iteration.

| Iteration | Model | SFT | RL | Geo-Sub-Aux |
|-----------|-------|-----|-----|-------------|
| | $\pi_\theta^0$ | ✔ | | 45.4 |
| First | $\pi_\theta^0$ | | ✔ | 44.7 |
| | $\pi_\theta^0$ | ✔ | ✔ | 46.9 |

Table 13: Comparison of error-rates on mathmatical function.

| Methods | Data Amount | FunctionQA |
|---------|-------------|------------|
| Qwen2-VL-7B | - | 58.0 |
| Fullset | 1.0k | 69.0 |
| second-0.3 | 0.7k | 72.0 |
| second-0.6 | 0.6k | 71.0 |
| second-1.0 | 0.4k | 68.0 |

As shown in Table 11, we additionally provide the results of Qwen2-VL-2B/7B on the MathVerse metric (Zhang et al., 2024a). It can be observed that the model's performance across the three iterations remains consistent with the previous metrics (Note: To ensure consistency across all experimental settings, we re-evaluated the Qwen2-VL model under a unified experimental protocol. Consequently, some results may differ from those reported in the original paper. For instance, the performance of Qwen2-VL-7B on MathVerse is 30.2, compared to the originally reported 32.5. Nevertheless, our final results still surpass the baseline by achieving a score of 34.9.).

### A.4 MORE ABLATION RESULTS.

#### A.4.1 THE INFLUENCE OF DIFFERENT ITERATION STRATEGIES.

In this main paper, we present a comparison of different iteration strategies across the second and third iterations. To provide a more comprehensive evaluation, we further conduct experiments on different strategies in the first iteration, as shown in Table 12. Under the same parameter settings, the results show that SFT+RL achieves the best performance in the first iteration, followed by SFT alone, while using RL-only training yields the least favorable outcomes.

#### A.4.2 GENERALIZATION OF THE ERROR-RATE MECHANISM.

Regarding the error rate, we have demonstrated its robustness across models of different scales in Section 4.2. In Table 13, we further present its generalization performance across diverse tasks.

#### A.4.3 ABALATION WITHOUT REASONING PROCESS.

Additionally, we have included results of training without reasoning data, as shown in the Table 14.

#### A.4.4 PERCENTAGE OF DISCARDED SYNTHETIC QUESTIONS.

Table 14: Results of training without reasoning data.

| Methods | Data Amount | Geo-Sub-Aux |
|---------|-------------|-------------|
| Qwen2-VL-7B | - | 40.4 |
| $C^2$-Evo-$1^{st}$ | 0.6k | 44.7 |
| $C^2$-Evo-$2^{nd}$ | 0.6k | 46.6 |

In the first round of question generation, 3,797 questions were produced. After answer-based filtering, 2,631 were retained (69.3%). A subsequent filtering step based on image consistency reduced the set to 2,539, from which 648 data samples were ultimately selected. In the second round, 720 questions were generated, 543 of which passed answer filtering (75.4%), and 384 were ultimately selected.

#### A.4.5 THE BREAKDOWN OF THE MATHVISTA TEST SET.

Table 15 provides a breakdown of the MathVista test set. This benchmark includes diverse domains such as Natural Images, Charts, and Scientific Figures, etc.

Table 15: Detailed performance breakdown on different visual domains.

| Model | Natural_Image | Abstract_Scene | Map_Chart | Scientific_Figure | Logical_Reasoning |
|---|---|---|---|---|---|
| Qwen2-VL-7B | 0.28 | 0.52 | 0.88 | 0.54 | 0.16 |
| **C2-Evo** | **0.32** | **0.64** | **0.96** | **0.59** | **0.22** |
| *Improvement* | +14.3% | +23.1% | +9.1% | +9.3% | +37.5% |

### A.4.6 COMPARISON WITH OTHER METHODS.

Table 16: Performance comparison with state-of-the-art models.

| Methods | Data Amount | Geo-Sub-Aux | MathVista |
|---|---|---|---|
| R1-VL-7B | 270K | - | 63.6 |
| R1-Onevision-7B | 165K | - | 64.1 |
| VLAA-Thinker-7B | 150K | - | 70.0 |
| OpenVLThinker-7B | 12K | - | 72.3 |
| Qwen2.5-VL-7B | - | 60.4 | 68.7 |
| C2-Evo-1st | 0.6K | 68.0 | 70.7 |
| C2-Evo-2nd | 0.6K | 76.8 | 71.1 |
| C2-Evo-3rd | 0.4K | **81.1** | **73.2** |
| *Improvement* | - | 34.2% | 7% |

We present experimental results based on the Qwen2.5-VL-7B backbone in Table 16. The results from R1-VL to OpenVLThinker are directly cited from the OpenVLThinker paper, while all other comparative results were obtained under identical experimental conditions to ensure a fair evaluation.

### A.4.7 ABLATION ON SINGLE PRINCIPLE.

Table 17: Ablation study on different principles.

| Principle | Geo-Sub-Aux |
|---|---|
| Constraints(0.2k) | 47.2 |
| New Theorems(0.1k) | 46.9 |
| Backward(0.2k) | 47.3 |
| C2-Evo-2nd | 48.0 |

In Table 17, we provide the second-round experimental results using a single criterion. Both the tabulated results and our preliminary experimental findings indicate that data diversity is critical for performance improvement.

## A.5 ADDITIONAL REMARKS.

### A.5.1 LIMITATIONS.

The experiments in this paper are primarily based on the Qwen model, and the proposed method has not been comprehensively validated on a wider range of models.

### A.5.2 FUTURE WORK.

We will intend to investigate the impact of model scale on performance, aiming to understand how architectural capacity interacts with evolutionary training dynamics.

**Question**:

c / b

a=14, b=48, and c=50, find cos A.

**Answer**:

0.96

**C²-Evo-2**:
*No auxiliary lines are needed, we can directly calculate cos A.*
*Reasoning Steps:*
*Step1: cosA = $(b^2 + c^2 - a^2)$ / (2bc).*
*Answer: 0.96*

**C²-Evo-3**:

*No auxiliary lines needed.*
*Reasoning Steps:*
*Step1: In a right triangle, cosA = $(b^2 + c^2 - a^2)$ / (2bc).*
*Answer: 0.96*

**C²-Evo-1**:
No auxiliary lines are needed. To find cos A, we can use the properties of the right triangle and the given lengths. Since triangle ABC is a right triangle with angle C being the right angle, we can use the Pythagorean to verify the triangle's propeties and calculate cos A.
Reasoning Steps:
Step1: cos A is calculated using the lengths of the sides.
Step2: Given a=14, b=48, and c=50, we find that cosA = $(b^2 + c^2 - a^2)$ / (2bc).
Step3: Plugging in the values, we get cos A = 0.96.
Answer: 0.96

Figure 8: **Changes in model outputs** across three testing iterations.

### A.5.3    THE USE OF LARGE LANGUAGE MODELS

In Section 3, we provide a detailed description of the integration of large language models, including the specific models employed and their usage. Additionally, during the writing process, we leveraged large language models for sentence polishing and grammatical refinement.

### A.6    MORE VISUAL RESULTS.

### A.6.1    EVOLUTION OF MODEL OUTPUTS ACROSS THREE TRAINING ITERATIONS.

Additionally, we present the model outputs for the same question solved at the end of each of the three iterations, as shown Figure 8.

### A.6.2    DATA DIFFICULTIES ACROSS THREE ITERATIONS.

As shown in Figure 9, we also evaluate the difficulty of training data from previous iterations using the model training in the preceding iteration. Our observations are as follows: 1) In the evaluation of the second iteration, the first iteration model $\pi^1_{\theta,\text{RL}}$ achieved completely wrong predictions on approximately $8\%$ of the data from the first iteration. 2) Despite the fact that the data generated in the second iteration was more challenging than that of the first iteration (*cf.* , as evidenced by the longer reasoning length in the Figure 3, the first iteration model $\pi^1_{\theta,\text{RL}}$ was still able to solve about $30\%$ of the problems from the second iteration. 3) In the evaluation of the third iteration, when the second iteration model $\pi^2_{\theta,\text{RL}}$ was used to evaluate the data from the first iteration, there is an increase in completely wrong predictions and a decrease in fully correct ones. This suggests that the model begins to exhibit *catastrophic forgetting*. 4) Even within the second iteration data, around $30\%$ of the questions are completely wrong ,which aligns with the assumption stated in the main text that the data difficulty increases across iterations.

### A.6.3    PROMPTS

All prompts used in our experiments are presented below. More detailed prompt designs will be included in the code.

Initial Prompt + Request (Differences between SKETCHPAD)

Here are some tools that can help you. All are python codes. They are in tools.py and will be imported for you.
Notice that The upper left corner of the image is the origin (0, 0). To implement the axis inversion so that the coordinate system starts from the top left corner, add the following code before plt.show() at the end of your program:
```
# Reverse the axis, starting from the top left
ax = plt.gca()
ax.xaxis.set_ticks_position('top')
ax.invert_yaxis()
def find_perpendicular_intersection(A, B, C):
    # Convert coordinates to numpy arrays for easier computation
    A = np.array(A)
    B = np.array(B)
    C = np.array(C)

    # Calculate the direction vector of line BC
    BC = C - B

    # Compute the slope of BC if not vertical
    if BC[0] != 0:
        slope_BC = BC[1] / BC[0]
```

```
      # Slope of the perpendicular line from A to BC
          slope_perpendicular = -1 / slope_BC
      else:
          # If line BC is vertical, then perpendicular line is horizontal
          slope_perpendicular = 0

      # Calculate the equation of the line passing through A and perpendicular to BC
      # y - y_A = slope_perpendicular * (x - x_A)
      # Rearrange to standard form Ax + By + C = 0
      if BC[0] != 0:
          A_coeff = -slope_perpendicular
          B_coeff = 1
          C_coeff = -A_coeff * A[0] - B_coeff * A[1]
      else:
          # If BC is vertical, AE must be horizontal
          A_coeff = 1
          B_coeff = 0
          C_coeff = -A[0]

      # Equation of line BC: (y - y_B) = slope_BC * (x - x_B)
      # Convert to Ax + By + C = 0 for line intersection calculation
      if BC[0] != 0:
          A_BC = -slope_BC
          B_BC = 1

          B_BC = 0
          C_BC = -B[0]

      # Solve the linear system of equations representing the two lines
      # [A_coeff B_coeff] [x] = [-C_coeff]
      # [A_BC    B_BC   ] [y]   [-C_BC   ]
      matrix = np.array([[A_coeff, B_coeff], [A_BC, B_BC]])
      constants = np.array([-C_coeff, -C_BC])

      # Use numpy to solve the linear system
      intersection = np.linalg.solve(matrix, constants)
      return intersection.tolist()

```

∘ ∘ ∘ ∘ ∘ ∘

# REQUIREMENTS #:
1. The generated actions can resolve the given user request # USER REQUEST # perfectly. The
user request is reasonable and can be solved. Try your best to solve the request.
2. If you think you can get the answer, please explains your reasoning step by step until you
can give the final answer.
3. Here's how the output format should look:

THOUGHT 0: [Provide your problem-solving method and whether you need to add any additional
auxiliary lines.]
ACTION 0: [Provide the matplotlib code you need to add auxiliary lines.]

OBSERVATION: Execution success. The output is as follows:
<the image outputs of the previous code is here.>

If you can get the answer, please reasoning step by step until you can give the final answer.
REASONING STEPS 0: [Provide a chain-of-thought, logical explanation of the problem. This should
outline step-by-step reasoning.]
ANSWER 0: [State the final answer in a clear and direct format. It must match the correct
answer exactly.]

Otherwise, please generate the next THOUGHT and ACTION.
THOUGHT 1:
ACTION 1:

REASONING STEPS 1:
ANSWER 1:

Now please generate only THOUGHT 0 and ACTION 0 in RESULT. If no action needed, also reasoning
step by step following Instructions below until you can give the final ANSWER: <REASONING
STEPS> <ANSWER> and ends with TERMINATE in the RESULT:\n# RESULT #:\n
        # Instructions #:

(1). Ensure your output is a single atomic reasoning step, which should be small and focused.
(2). Ensure that your reasoning incorporates all relevant details from the provided image.
(3). Break down your explanation into clear, concise steps. Use as many reasoning steps as possible while avoiding unnecessary or redundant information.
(4). In your reasoning process, utilize various approaches to explore the answer comprehensively, ensuring a thorough analysis.
(5). Base your reasoning strictly on the information available in the image and prior context to prevent inaccuracies.

REASONING STEPS: [Provide a chain-of-thought, logical explanation of the problem. This should outline step-by-step reasoning.] Step1: We need to .... . Step2: To find the ....
ANSWER: [The obtained ANSWER should be simple and correct"]

## Sub-Problem Generation

Treat follow detailed description as an image: {responses}.
QUESTION STYPLE EXAMPLES: \n
1.In \\odot O, E C and A B are diameters, and \\angle B O D \\cong \\angle D O E \\cong \\angle E O F \\cong \\angle F O A. Find m \\widehat C B F.\n
2.Quadrilateral A B C D is inscribed in \\odot Z such that m \\angle B Z A = 104, m \\widehat C B = 94, and A B \\parallel D C. Find m \\angle B D A.\n
3.isosceles trapezoid T W Y Z with \\angle Z \\cong \\angle Y, m \\angle Z = 30 x, \\angle T \\cong \\angle W, and m \\angle T = 20 x, find Z.\n
4.Find the area of the shaded region. Assume that all polygons that appear to be regular are regular. Round to the nearest tenth.\n
5.For trapezoid J K L M, A and B are midpoints of the legs. If A B = 57 and K L = 21, find J M.\n

PRINCIPLES:
1. Utilize the given geometric relationships.
2. Apply some new theorems or conditions.
3. Reverse the reasoning process to derive new problems.

Your task is to imagine you are looking at the given picture, and based on the style of the QUESTIONS STYLE EXAMPLES, PRINCIPLES, REASONING STEPS, generate 4 to 10 new questions, ensuring each sub-question is diverse, correct, solvable, and rigorous.
Ensure that each sub-question has sufficient conditions to be solved independently, without relying on the detailed description.
Each question should be logically reasoned through to arrive at the final answer.

Input:

Formal Description (responses), Reasoning Steps
Output:
Question: ...
Answer: ...

## Challenging-Problem Generation

Based on the subproblems {sim_problem}, Formal Description, compare the differences and connections between geometric shapes in these subproblems, and combine these geometric shapes to generate 5 to 12 new complex questions , ensuring each sub-question is complex, diverse, correct, solvable, and rigorous.
Ensure that the problem strictly adheres to the description in the diagram: {responses}.
To ensure each question is complex and fully utilizes these geometric shapes.
Ensure each problem utilizes geometric theorems.
Ensure that each sub-question has sufficient conditions to be solved independently, without relying on the detailed description.
Each question should be logically reasoned through to arrive at the final answer.
Only provide the final answer, without showing the intermediate reasoning process.
The final output should follow this format:
Question: ...
Answer: ...

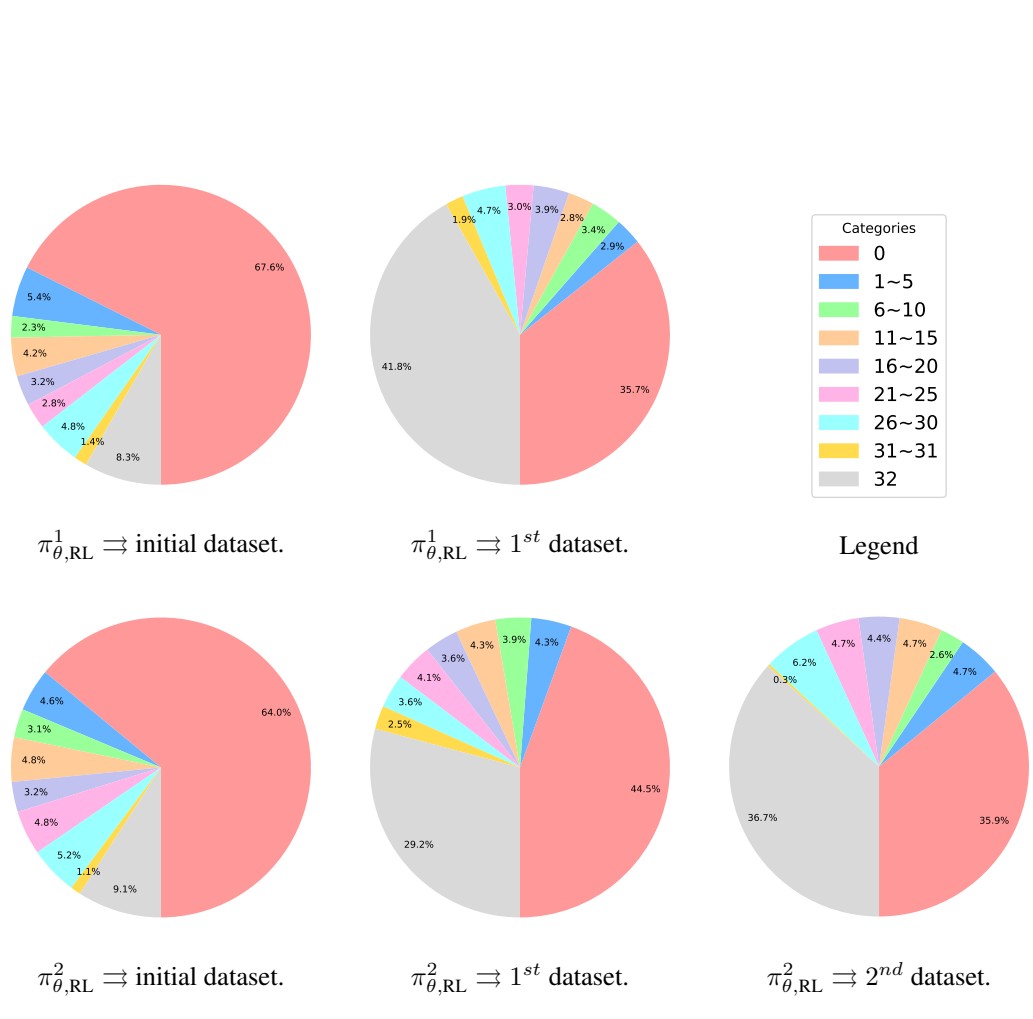

Figure 9: **Evolution of data difficulty across iterations (assessed by error-rate).** $\pi_\theta \rightrightarrows \mathcal{D}$ denotes the difficulty distribtions of dataset $\mathcal{D}$ evaluated using model $\pi_\theta$ based on error-rate.

