# OpenReview forum: "C2-Evo: Co-Evolving Multimodal Data and Model for  Self-Improving Reasoning"
_ICLR.cc/2026/Conference — Submitted to ICLR 2026_

### Official Review · Reviewer_bnAy · 2025-10-25

**Soundness:** 2
**Presentation:** 3
**Contribution:** 2
**Rating:** 4
**Confidence:** 4

**Summary:**

This paper proposes C2-Evo, a closed-loop framework that jointly evolves training data and model capabilities. C2-Evo integrates cross-modal data evolution with adaptive model training. Experiments on several mathematical reasoning benchmarks show that C2-Evo yields considerable performance gains without external data.

**Strengths:**

• This paper is well-organized and easy to read.

• Developing self-improving reasoning capabilities in MLLMs is an important and interesting task.

• The proposed C2-Evo is reasonable and can be applied to various fine-tuning scenarios.

**Weaknesses:**

• The proposed method is incremental, combining data-evolving and model-evolving methods. In particular, the co-evolving data and model method merely applies widely adopted SFT and GRPO with accuracy reward and format reward to fine-tune the MLLMs.

• It would be better to include some advanced data-evolving and method-evolving baselines. It would be unfair to compare the original LLMs (Qwen2-VL).

**Questions:**

It would be better to include some advanced data-evolving and method-evolving baselines. It would be unfair to compare the original LLMs (Qwen2-VL).

---

> ### Author Response · Authors · 2025-11-24
>
> > W1. The proposed method is incremental, combining data-evolving and model-evolving methods. In particular, the co-evolving data and model method merely applies widely adopted SFT and GRPO with accuracy reward and format reward to fine-tune the MLLMs.
>
> We clarify that the **core innovation of C2-Evo lies not in the underlying training algorithms, but in the holistic design of the evolutionary closed-loop**. This contribution is recognized by other reviewers, who acknowledged our innovation in "automatically align task difficulty with model competence," noting that the framework design "highlight the core contributions effectively" and addresses "an important gap in MLLM self-improvement research."
> To address this:
>
> **1. Synchronized Visual-Textual Evolution.** Existing methods often suffer from decoupled difficulty scaling. For instance, R-CoT lacks the combinatorial complexity of visual elements, while Mind-GYM relies on fixed visual components, leading to a difficulty plateau in later stages. In contrast, C2-Evo ensures continuous evolution by:
> **Increasing visual complexity via auxiliary lines:** Addressing the poor accuracy of auxiliary line drawing in previous works, we introduce a coordinate-based generation method to ensure precision.
> **Scaling textual complexity:** Via our sub-problem and complex question synthesis mechanism, we ensure that the questions fully mine the semantic information within the images. This synergy guarantees that both modalities evolve synchronously, preventing data difficulty from plateauing after two or three rounds of evolution.
>
> **2. Co-evolution of Data Difficulty and Model Capability.** Unlike approaches such as OpenVLthinker, which rely on manual difficulty design, C2-Evo employs an **error-rate-based filtering mechanism** to automatically align data difficulty with the model's current competence. This adaptive alignment results in exceptional data efficiency. Notably, our method achieves performance comparable to R1-VL (trained on 120k samples) while using less than 1k samples.
>
> C2-Evo is a cohesive framework rather than a simple incremental improvement. Compared to self-improvement paradigms like ViLA$^2$ and MMEvol, which encounter challenges such as symbolic/formulaic errors and data redundancy in high-order reasoning, the design of the C2-Evo framework effectively addresses these issues, representing a superior methodology for **high-order and complex reasoning tasks**.
>
>
> > W2.  It would be better to include some advanced data-evolving and method-evolving baselines. It would be unfair to compare the original LLMs (Qwen2-VL).
>
> We have provided comprehensive comparisons to validate our approach:
>
> Table 2: Compares our method against R1-VL, a baseline integrating SFT and Reinforcement Learning (RL).
>
> Table 8: Compares our framework with OpenVLThinker, a representative approach for synchronized data-model evolution.
>
> Table 9: Presents an ablation study contrasting text-only evolution with our proposed joint text-image evolution, demonstrating the necessity of the latter.

---

### Official Review · Reviewer_KjBx · 2025-10-28

**Soundness:** 2
**Presentation:** 3
**Contribution:** 2
**Rating:** 4
**Confidence:** 5

**Summary:**

This paper introduces **C2-Evo**, a self-improving framework designed to jointly enhance multimodal large language models (MLLMs) and their training data. The framework addresses two key challenges in existing methods: mismatched visual-textual data complexities and the separation of data evolution from model improvement.

C2-Evo operates through two integrated loops: (i) a cross-modal data evolution loop that generates complex multimodal problems with balanced challenges, and (ii) a data-model evolution loop that adapts training tasks based on model performance. Through these continuous iterations, C2-Evo achieves notable improvements across mathematical reasoning benchmarks.

**Strengths:**

1. **Well-Presented and Clear**: The paper is written in a clear and accessible manner, ensuring that readers can easily understand, reproduce, and validate the proposed methods.

2. **Unified Evolution of Vision and Text**: The paper addresses the critical challenge of integrating the evolution of visual and textual data into a unified process, allowing simultaneous evolution of images and instruction texts. This significantly enhances the diversity of the dataset. As image evolution is inherently challenging, the authors wisely focus on subcategories like mathematical multimodal question answering, which are easier to implement via code modifications while highlighting the core contributions effectively.

3. **Use of SFT and Advanced Post-Training Techniques**: In addition to supervised fine-tuning (SFT), the paper employs GRPO and other post-training techniques to improve model performance in targeted domains such as mathematics. This not only aligns with current research trends but also helps deliver further performance enhancements in these specialized subfields.

**Weaknesses:**

1. **Overclaiming the Contribution of Cyclic Evolution**: While the paper introduces the concept of cyclic evolution between data and models under the name C2-Evo, this approach is already common in prior works [1,2]. The lack of discussion around these related classic studies may lead to an overstatement of its contribution and novelty.

2. **Use of Outdated Base Models**: The experiments are conducted using Qwen2 series models, despite the availability of the newer Qwen3 series models. Results obtained using older models may not be fully convincing. It is necessary to validate the method with at least the Qwen2.5 series, which could better assess the timeliness and adaptability of the proposed data evolution mechanism.

3. **Challenges in Image Evolution and Its Necessity**: While image evolution is acknowledged as a difficult task, even with narrowed subcategories such as mathematics, the diversity of evolved images may still be limited, and there is a risk of introducing errors. This raises doubts about the necessity of performing image evolution. Ablation studies focusing solely on evolving textual instructions (without image synchronization) are necessary to provide evidence for the added value of image evolution, but the paper does not include such experiments.

[1]  MMEvol: Empowering Multimodal Large Language Models with Evol-Instruct

[2]  VILA^2: VILA Augmented VILA

**Questions:**

1. **Similarity to Existing Works Without Explicit Comparison**: The presentation style of this work, including the shift in reasoning step distribution and the complexity analysis of instruction data, closely resembles the MMEvol [1] study. However, the authors have not directly compared their method to MMEvol or clarified the distinctive contributions of their approach. A detailed classification and differentiation of their core contributions are needed to establish novelty.

2. **Questionable Necessity and Generality of Image Evolution**: While the paper focuses on multimodal evolution, it fails to prove the necessity of image evolution. Furthermore, no experiments were conducted on general multimodal datasets, raising doubts about the generality and utility of the approach. When scaled to a larger dataset size, such as 10M samples for large-scale training, the evolved data may struggle to significantly enhance multimodal model performance. For instance, while MMEvol shows notable improvement on small datasets, its impact diminishes significantly when scaled to 10M data points (e.g., Mammthvl [2]). This could suggest a similar scalability limitation for C2-Evo.

3. **Ambiguous Claim of "Self-Improving"**: The paper casually claims to achieve self-improving capabilities, but it still relies heavily on advanced multimodal models like GPT-4o for assistance, making this more akin to a form of data distillation rather than true self-improvement. Methods like ViLA^2 [3], which more rigorously align with self-improving frameworks, might be more fitting representatives of this paradigm. Further clarification and justification of C2-Evo's self-improvement claim are necessary.

[1]  MMEvol: Empowering Multimodal Large Language Models with Evol-Instruct

[2] MAmmoTH-VL: Eliciting Multimodal Reasoning with Instruction Tuning at Scale

[3]  VILA^2: VILA Augmented VILA

---

> ### Author Response · Authors · 2025-11-24
>
> > W1. Overclaiming the Contribution of Cyclic Evolution: While the paper introduces the concept of cyclic evolution between data and models under the name C2-Evo, this approach is already common in prior works [1,2]. The lack of discussion around these related classic studies may lead to an overstatement of its contribution and novelty.
>
>
> We thank you for bringing these relevant works to our attention. We have **included** them in the "Related Work" section (**L140, L601, L582**). Compared to these approaches, the novelty of C2-Evo is highlighted in two key aspects:
>
> **1. Synchronized Difficulty Evolution.** MMEvol and MAmmoTH-VL primarily focus on mining deeper visual semantics to evolve textual questions, similar to the "fixed-source evolution" approach (e.g., MindGYM). Relying on **fixed visual elements inevitably leads to a plateau in reasoning difficulty during later stages**. In contrast, C2-Evo simultaneously evolves both visual and textual components, preventing this bottleneck and ensuring consistent difficulty scaling across modalities.
>
> **2. Alignment of Data Complexity with Model Capability.** While these works rely on scaling up data volume (e.g., 163K or 12M samples), our approach prioritizes increasing data complexity while dynamically aligning task difficulty with the model’s current competence. This strategy results in significantly higher data efficiency compared to massive-scale generation methods.
>
> > W2. Use of Outdated Base Models: The experiments are conducted using Qwen2 series models, despite the availability of the newer Qwen3 series models. Results obtained using older models may not be fully convincing. It is necessary to validate the method with at least the Qwen2.5 series, which could better assess the timeliness and adaptability of the proposed data evolution mechanism.
>
> We present experimental results based on the **Qwen2.5-VL-7B** backbone. The results from R1-VL to OpenVLThinker are directly cited from the OpenVLThinker paper, while all other comparative results were obtained under identical experimental conditions to ensure a fair evaluation. For detailed hyperparameter configurations and settings, please refer to Section 4.1 of the main paper. These results have been **added to Table 16 in Appendix.**
>
> | Methods | Data Amount | Geo-Sub-Aux | MathVista
> | -------- | -------- | -------- |-------- |
> | R1-VL-7B   | 270K  |  -    | 63.6     |
> | R1-Onevision-7B | 165K    |   -   | 64.1     |
> | VLAA-Thinker-7B  | 150K   |  -    | 70.0    |
> | OpenVLThinker-7B | 12K    |  -    | 72.3     |
> | Qwen2.5-VL-7B  | -  | 60.4     | 68.7     |
> | C2-Evo-1st | 0.6K    | 68.0    |   70.7   |
> | C2-Evo-2nd | 0.6K    |   76.8   |   71.1   |
> | C2-Evo-3rd | 0.4K    |  **81.1**    |   **73.2**   |
> | *Improvement* | -    |    34.2%  |  7.0%    |

---

> ### Author Response · Authors · 2025-11-24
>
> > W3. Challenges in Image Evolution and Its Necessity: While image evolution is acknowledged as a difficult task, even with narrowed subcategories such as mathematics, the diversity of evolved images may still be limited, and there is a risk of introducing errors. This raises doubts about the necessity of performing image evolution. Ablation studies focusing solely on evolving textual instructions (without image synchronization) are necessary to provide evidence for the added value of image evolution, but the paper does not include such experiments.
>
> We appreciate your insightful comment, which touches upon a critical aspect validated through our extensive experiments. We address the robustness and necessity of image evolution as follows:
>
> **1. Challenges and Robustness.** We evaluated various models for this task. While open-source models (e.g., Qwen2.5-VL) are prone to errors, stronger models like GPT-4o demonstrate superior stability. Crucially, we adopted the **coordinate-based generation method (L210)**. Providing explicit coordinates for code-based rendering significantly minimizes errors. Furthermore, a certain degree of diversity is maintained as the model occasionally introduces randomness when generating auxiliary lines. While we explored adding random noise (e.g., random points/lines), preliminary results showed marginal impact on early evolution, although this strategy may prove more beneficial for extensive future iterations. Additionally, this design is generalizable to other tasks with structured and modular characteristics.
>
> **2. Necessity of Image Evolution.** Relying on a **fixed set of visual elements inevitably leads to a difficulty plateau in later stages**. Image evolution is therefore essential to sustain difficulty scaling. This is empirically supported by our ablation study in **Table 9**, where joint text-image evolution consistently outperforms text-only evolution, confirming the benefit of synchronized cross-modal improvement.
>
>
> > W4. Similarity to Existing Works Without Explicit Comparison: The presentation style of this work, including the shift in reasoning step distribution and the complexity analysis of instruction data, closely resembles the MMEvol [1] study. However, the authors have not directly compared their method to MMEvol or clarified the distinctive contributions of their approach. A detailed classification and differentiation of their core contributions are needed to establish novelty.
>
> See W1.

---

> ### Author Response · Authors · 2025-11-24
>
> > W5. Questionable Necessity and Generality of Image Evolution: While the paper focuses on multimodal evolution, it fails to prove the necessity of image evolution. Furthermore, no experiments were conducted on general multimodal datasets, raising doubts about the generality and utility of the approach. When scaled to a larger dataset size, such as 10M samples for large-scale training, the evolved data may struggle to significantly enhance multimodal model performance. For instance, while MMEvol shows notable improvement on small datasets, its impact diminishes significantly when scaled to 10M data points (e.g., Mammthvl [2]). This could suggest a similar scalability limitation for C2-Evo.
>
>
> Response regarding Image Evolution, Task Practicality, and Data Scale
>
> **1. Necessity of Image Evolution.** Please refer to our detailed response in W3.
>
> **2. Generalization on Broad Reasoning Benchmarks.** Beyond geometry, C2-Evo demonstrates consistent superiority on general multimodal reasoning benchmarks such as **MMMU, MME, and MathVista** (e.g., Table 2, 7). To specifically address the "applicability" concern, we provide a breakdown of the MathVista test set below. This benchmark includes diverse domains such as **Natural Images, Charts, and Scientific Figures**, which are distinct from geometry problems. These results have been **added to Table 15 in Appendix.**
>
> | Model | Natural_Image |  Abstract_Scene | Map_Chart | Scientific_Figure |  Logical_Reasoning
> | -------- | -------- | -------- |  -------- |-------- |-------- |
> | Qwen2-VL-7B     | 0.28  |   0.52  | 0.88     |  0.54       | 0.16
> | **C2-Evo**     | **0.32**    |  **0.64**     | **0.96**     |  **0.59**     | **0.22**
> | *Improvement*  | +14.3%  |  +23.1%       | +9.1%     | +9.3%    | +37.5%
>
> As shown, C2-Evo achieves consistent improvements across all domains. Notably:
> **Real-world Interpretation:** The gains in Map Chart (+9.1%) and Scientific Figure (+9.3%) verify the model's ability to interpret complex schematic information independent of geometric theorems.
> **General Perception:** The +14.3% gain in Natural Images indicates that our "visual evolution" mechanism effectively enhances the perception of complex details in real-world photos, proving its applicability to general visual scenarios.
>
> **3. Clarification on Scope.** Our framework targets schematic and reasoning-heavy scenarios (e.g., charts, scientific reasoning, logical reasoning) rather than simple recognition tasks (e.g., "a cat on the grass"). The results above confirm that C2-Evo is robust across these complex domains, validating its applicability beyond pure mathematics. While **ViLA$^2$** is a representative self-improvement method, its application is primarily limited to basic tasks such as VQA. When generalizing this paradigm to high-order reasoning, it often encounters significant challenges, including symbol errors and formula incoherence. These inaccuracies trigger **error accumulation** during the iterative process, leading to evolutionary failure in complex scenarios. In contrast, our framework offers a robust paradigm optimized for high-order reasoning, effectively mitigating these issues.
>
> **4. Expansion of Data Scale.** We clarify that C2-Evo fundamentally differs from MMEvol and Mammoth-VL in its evolution mechanism. Unlike these approaches, C2-Evo dynamically screens for datasets specifically adapted to the model's current capability (difficulty alignment). This process effectively filters out a vast amount of redundant, low-difficulty samples. Even if we scale up, our complexity-based mechanism avoids the inefficiency of generating massive amounts of candidate data only to filter them out. By selectively targeting data difficulty, the framework bypasses the creation of redundant, low-value samples. Consequently, the objective of our multimodal evolution is to scale data complexity rather than mere quantity. Notably, C2-Evo achieves performance comparable to R1-VL while using less than 1% of the data volume (0.6k vs. 120k), demonstrating the framework's exceptional data efficiency.

---

> ### Author Response · Authors · 2025-11-24
>
> > W6. Ambiguous Claim of "Self-Improving": The paper casually claims to achieve self-improving capabilities, but it still relies heavily on advanced multimodal models like GPT-4o for assistance, making this more akin to a form of data distillation rather than true self-improvement. Methods like ViLA^2 [3], which more rigorously align with self-improving frameworks, might be more fitting representatives of this paradigm. Further clarification and justification of C2-Evo's self-improvement claim are necessary.
>
> We thank you for your expertise and for highlighting this relevant work. We have included it in the "Related Work" section (L582). We clarify the distinctions of our method from two perspectives:
>
> **1. Distinction from Simple Distillation.** C2-Evo is a **dynamic evolutionary framework**, fundamentally different from static data distillation. In our approach, GPT-4o serves merely as a base foundation model for generation. The core contribution lies in the synchronized co-evolution of visual and textual modalities, combined with a mechanism that dynamically aligns data difficulty with model competence. The error-based filtering ensures this adaptive alignment, distinguishing our iterative framework from simple data distillation methods.
>
> **2. Common Practice of Model-Based Data Generation.** We clarify that leveraging advanced closed-source models (e.g., GPT-4o) for data synthesis is a widely adopted methodology in recent research. This practice is standard in works such as DeepSeek-R1, LLaVA-CoT, and R1-OneVision. Particularly in the domain of iterative evolution, representative studies like **MMEvol** also rely on such models for data generation.
>
> **3. ViLA$^2$: Limited Generalization to High-Order Reasoning.** While ViLA$^2$ presents an effective paradigm relying on a model's intrinsic capabilities, complex high-order reasoning (e.g., mathematics) imposes significantly higher demands than simple VQA tasks. In our preliminary experiments, we evaluated various models for question generation, including Qwen2.5-VL-32B/72B. We found that these open-source models frequently introduced symbolic and formulaic errors, which accumulated and led to failure in subsequent evolution stages. Applying a question generation paradigm (like ViLA$^2$) using smaller models in complex high-order reasoning tasks would further exacerbate these errors. Consequently, C2-Evo strategically leverages GPT-4o to minimize initial inaccuracies, while our framework's design effectively mitigates the error propagation issues often encountered in question generation for high-complexity tasks.

---

### Official Review · Reviewer_j9to · 2025-10-29

**Soundness:** 3
**Presentation:** 3
**Contribution:** 2
**Rating:** 4
**Confidence:** 4

**Summary:**

The paper proposes C2-Evo, a self-improving multimodal reasoning framework designed to address two key challenges in current Multimodal Large Language Models (MLLMs): (1) the mismatch between visual and textual data complexity, and (2) the misalignment between model capability and task difficulty. C2-Evo introduces two closed-loop co-evolution mechanisms: a cross-modal data co evolution loop, which generates semantically consistent and progressively more complex visual-textual reasoning problems (e.g., geometric diagrams, mathematical functions) using GPT-4o; and a data–model co-evolution loop, which dynamically selects training samples based on model error rates and alternates between Supervised Fine-Tuning (SFT) and Group Relative Policy Optimization (GRPO). Experiments on benchmarks such as Geo-Sub, MathVista, and MathVerse show consistent improvement across three iterations. Notably, C2-Evo achieves performance close to GPT-4o while using less than 1% of the training data, outperforming strong open-source baselines like Qwen2-VL.

**Strengths:**

1. The use of error-rate filtering and alternating SFT/GRPO training enables the model to automatically align task difficulty with model competence.
2. Demonstrates substantial performance gains and shows cross-task generalization on multiple benchmarks.

**Weaknesses:**

1. The first claim lacks sufficient examples or statistical justification and appears somewhat counterintuitive. Existing works on multimodal self-evolution typically revolve around data evolution and iteration based on image semantics, where visual and textual modalities are mutually complementary. However, the paper states that “Prior methods often address visual and textual components in isolation,” which is not very convincing and precise.
2. Although a model-based error filtering mechanism is introduced, there will still inevitably be residual noisy data. How does the framework further ensure the reliability and cleanliness of the iteratively generated data, _e.g._, incorrect mathematical formulas, variable confusion, or hallucinations, that appear during the iterative process?
3. The title emphasizes “self-improving reasoning on multimodal data,” but in practice, the work only addresses diagram-based mathematical reasoning tasks, with a relatively small dataset scale. Moreover, when scaled up, is the framework still efficient and cost-effective?

**Questions:**

Please see the Weaknesses.

---

> ### Author Response · Authors · 2025-11-24
>
> > W1. The first claim lacks sufficient examples or statistical justification and appears somewhat counterintuitive. Existing works on multimodal self-evolution typically revolve around data evolution and iteration based on image semantics, where visual and textual modalities are mutually complementary. However, the paper states that “Prior methods often address visual and textual components in isolation,” which is not very convincing and precise.
>
> We thank you for the correction suggestion. We clarify that our intended meaning of "isolation" specifically refers to the asynchrony in difficulty evolution between modalities.
>
> **1. Visual-Dominant Evolution (e.g., R-CoT):** While visual elements may become progressively richer, the accompanying textual questions often remain focused on local details. This approach frequently **neglects complex relationships among multiple elements**, causing textual difficulty to lag behind visual complexity.
>
> **2. Fixed-Source Evolution (e.g., MindGYM):** Although these methods extract richer semantics from images, relying on a fixed set of visual elements inevitably leads to **a difficulty plateau in later evolutionary stages**.
>
> In contrast, our framework synchronizes complexity growth across both modalities to prevent such misalignment. **We have revised the manuscript to include this detailed explanation (L081)**.
>
>
> > W2. Although a model-based error filtering mechanism is introduced, there will still inevitably be residual noisy data. How does the framework further ensure the reliability and cleanliness of the iteratively generated data, e.g., incorrect mathematical formulas, variable confusion, or hallucinations, that appear during the iterative process?
>
> We appreciate your insightful comment regarding data correctness, a critical aspect we extensively validated. We address the potential errors in generated Questions and Chain-of-Thought (CoT) through two main strategies:
>
> **1. Rigorous Model Selection.** We conducted preliminary experiments using various foundation models for data generation, including Qwen2.5-VL-32/72B, DeepSeek, and GPT-4o, etc. We observed that open-source models frequently introduced symbolic inconsistencies and formulaic errors, which led to failures in the subsequent evolution process. Consequently, we selected GPT-4o as our generator, as it demonstrated a significantly **lower initial error rate** in both Question formulation and CoT reasoning. Notably, while our experiments indicated that the combination of Doubao for formal description and DeepSeek for question generation yielded the optimal results, we ultimately prioritized GPT-4o to ensure broader accessibility and reproducibility for the research community, given potential access limitations to these specific APIs.
>
> **2. Multi-Stage Filtering.** To further mitigate residual errors, we implemented a multi-round filtering pipeline, incorporating both **answer-based verification and consistency checks** between visual variables and textual conditions (L275-L277). Our manual inspections and consistent performance improvements confirm that the impact of remaining errors is negligible. Detailed statistics on sample retention after these filtering steps are provided in Appendix A4.4.
>
>
> While we acknowledge that achieving 100% accuracy in synthetic data is challenging, exploring methods to learn from noisy data remains a promising direction for our future research.

---

> ### Author Response · Authors · 2025-11-24
>
> > W3. The title emphasizes “self-improving reasoning on multimodal data,” but in practice, the work only addresses diagram-based mathematical reasoning tasks, with a relatively small dataset scale. Moreover, when scaled up, is the framework still efficient and cost-effective?
>
> We address the points regarding task scope and dataset size as follows:
>
> **1. Generalization Beyond Mathematical Diagrams.** We provide two lines of evidence to justify the "multimodal" claim in our title:
>
> **Performance on General Benchmarks:** To support this beyond pure mathematics, we give our performance on **MMMU** (a benchmark covering 30+ disciplines) and **MME** (perception & cognition), where C2-Evo consistently outperforms the baseline (Table 2, 7).
>
> **Transferability Analysis:** Specifically on MathVista, which contains diverse visual contexts beyond geometry, C2-Evo shows robust transferability. As shown in the table below, we observe significant gains in **Natural Images (+14.3%) and Scientific Figures (+9.3%)**, proving that our framework enhances general capabilities, not just theorem application. These results have been **added to Table 15 in Appendix.**
>
> | Model | Natural_Image |  Abstract_Scene | Map_Chart | Scientific_Figure |  Logical_Reasoning
> | -------- | -------- | -------- |  -------- |-------- |-------- |
> | Qwen2-VL-7B     | 0.28  |   0.52  | 0.88     |  0.54       | 0.16
> | **C2-Evo**     | **0.32**    |  **0.64**     | **0.96**     |  **0.59**     | **0.22**
> | *Improvement*  | +14.3%  |  +23.1%       | +9.1%     | +9.3%    | +37.5%
>
>
> **2. Data Efficiency & Scalability.** We clarify that the compact dataset size is not a limitation but a demonstration of high data efficiency. Unlike approaches such as R1-VL that rely on massive data scaling, C2-Evo prioritizes **scaling data complexity rather than quantity** (e.g., the framework dynamically aligns data difficulty with the model’s evolving capability, effectively pruning large volumes of redundant samples with similar difficulty levels). This represents a more computationally efficient and scalable path for self-improvement. Notably, our method achieves experimental results comparable to R1-VL while utilizing less than 1% of the data volume (0.6k vs. 120k). Therefore, our design does not inherently require massive data expansion to be effective. Even in a hypothetical scale-up scenario, our **complexity-based pruning mechanism** would ensure cost-effectiveness by filtering out redundant samples, maintaining high efficiency.

---

### Official Review · Reviewer_DRDG · 2025-11-01

**Soundness:** 3
**Presentation:** 3
**Contribution:** 2
**Rating:** 6
**Confidence:** 3

**Summary:**

This paper proposes C2-Evo, a closed-loop self-improvement framework for MLLMs that jointly evolves both training data and model capabilities, with a focus on diagram-based mathematical reasoning. It addresses two key problems: the mismatch between visual and textual complexity in multimodal datasets, and the misalignment between task difficulty and model ability during iterative training.

The main contributions include: (i) a cross-modal data evolution loop generating progressively complex visual-text problems; (ii) a data-model evolution loop that adapts training difficulty to model performance with error-rate filtering; (iii) empirical experiments showing significant performance gains with minimal data; and (iv) plans to release code, models, and datasets. Multiple baseline comparisons are provided for open-source and closed-source reasoning models.

**Strengths:**

1. Introduces a joint co-evolution mechanism for both data and model, addressing an important gap in MLLM self-improvement research.
2. Demonstrates performance improvements across multiple reasoning benchmarks while requiring <1% of baseline training data.
3. Shows generalization beyond geometry, to mathematical function reasoning and broader multimodal benchmarks.

**Weaknesses:**

1. Evaluation primarily focuses on geometry problems, which may limit applicability claims to other complex multimodal domains.
2. Heavy reliance on GPT-4o as an oracle for data generation may reduce accessibility for researchers without closed-source model access.
3. Limited ablation on individual components of the framework (e.g., impact of each guiding principle in question generation) restricts the interpretability of improvements.

**Questions:**

1. How sensitive is C2-Evo’s performance to the choice of the oracle model, and can similar results be achieved with open-source or smaller models?
2. Could you provide a detailed ablation separating the contributions of visual complexity evolution vs. textual complexity evolution?
3. How does the computational cost of iterative evolution training compare to static dataset fine-tuning?
4. Missing reference: MMEvol: Empowering Multimodal Large Language Models with Evol-Instruct (ACL 2025)

---

> ### Author Response · Authors · 2025-11-24
>
> > W1. Evaluation primarily focuses on geometry problems, which may limit applicability claims to other complex multimodal domains
>
> We thank you for acknowledging our generalization capabilities in the "Strengths" section. We understand your concern regarding applicability. To address this:
>
> 1. **Generalization on Broad Reasoning Benchmarks.** Beyond geometry, C2-Evo demonstrates consistent superiority on general multimodal reasoning benchmarks such as **MMMU, MME, and MathVista** (e.g., Table 2, 7). To specifically address the "applicability" concern, we provide a breakdown of the MathVista test set below. This benchmark includes diverse domains such as **Natural Images, Charts, and Scientific Figures**, which are distinct from geometry problems. These results have been **added to Table 15 in Appendix.**
>
> | Model | Natural_Image |  Abstract_Scene | Map_Chart | Scientific_Figure |  Logical_Reasoning
> | -------- | -------- | -------- |  -------- |-------- |-------- |
> | Qwen2-VL-7B     | 0.28  |   0.52  | 0.88     |  0.54       | 0.16
> | **C2-Evo**     | **0.32**    |  **0.64**     | **0.96**     |  **0.59**     | **0.22**
> | *Improvement*  | +14.3%  |  +23.1%       | +9.1%     | +9.3%    | +37.5%
>
> As shown, C2-Evo achieves consistent improvements across all domains. Notably:
> **Real-world Interpretation:** The gains in Map Chart (+9.1%) and Scientific Figure (+9.3%) verify the model's ability to interpret complex schematic information independent of geometric theorems.
> **General Task:** The +14.3% gain in Natural Images indicates that our "visual evolution" mechanism effectively enhances the perception of complex details in real-world photos, proving its applicability to general visual scenarios.
>
> 2. **Mathematical reasoning represents a significant challenge in Visual Reasoning.** We chose geometry and mathematical functions not to limit our scope, but because they serve as typical representatives of high-order visual reasoning (requiring rigorous logic, fine-grained perception, and multi-step derivation). This aligns with the "Reasoning-Intensive" domain, which is distinct from "Recognition-Intensive" tasks (e.g., general captioning). Solving mathematical problems implies the capability to handle complex spatial relationships and logic, which is a challenge for current MLLMs.
>
> 3. **Clarification on Scope.** Our framework targets schematic and reasoning-heavy scenarios (e.g., charts, scientific reasoning, logical reasoning) rather than simple recognition tasks (e.g., "a cat on the grass"). The results above confirm that C2-Evo is robust across these complex domains, validating its applicability beyond pure geometry.
>
> > W2. Heavy reliance on GPT-4o as an oracle for data generation may reduce accessibility for researchers without closed-source model access.
>
>
> Our choice of data generation models is based on extensive empirical verification. We offer the following clarifications:
>
> **Necessity of High-Capability Models.** Mathematical reasoning tasks impose strict requirements on fine-grained perception and rigorous logic. In our preliminary experiments, we evaluated various models (including Qwen2.5-VL-32B/72B, DeepSeek, GPT4o, Doubao, etc.). We found that **current open-source models frequently exhibited hallucinations, such as symbolic errors and variable inconsistencies**. These inaccuracies led to severe error accumulation, causing failures in the evolution process. Consequently, we selected GPT-4o to ensure high-quality generation. Combined with our rigorous filtering strategies, the impact of error accumulation is negligible. We anticipate that as open-source VLM capabilities advance, they will eventually replace proprietary models in this pipeline.
>
>
> **Data Accessibility.** To mitigate concerns regarding the dependency on closed-source APIs, we commit to **open-sourcing** the synthesized dataset. This will allow the community to build upon our work without requiring access to proprietary models.

---

> ### Author Response · Authors · 2025-11-24
>
> > W3. Limited ablation on individual components of the framework (e.g., impact of each guiding principle in question generation) restricts the interpretability of improvements.
>
> We thank you for your in-depth understanding of our work. This is indeed an issue we also observed during our experiments.
> In our preliminary experiments, we observed that generating sub-problems without multi-criteria guidance resulted in significant redundancy and insufficient diversity. Consequently, **the multi-criteria mechanism was adopted specifically to ensure diverse question generation**. Nevertheless, we still provide the second-round experimental results using a single criterion in the table below. Both the tabulated results and our preliminary experimental findings indicate that data diversity is critical for performance improvement. These results have been **added to Table 17 in Appendix.**
>
> | Principle |  Geo-Sub-Aux |
> | -------- | -------- |
> | Constraints(0.2k)     |   47.2   |
> | New Theorems(0.1k)     |  46.9    |
> | Backward(0.2k)     |  47.3    |
> | C2-Evo-2nd     |  48.0    |
>
> > W4. How sensitive is C2-Evo’s performance to the choice of the oracle model, and can similar results be achieved with open-source or smaller models?
>
> See W2.
>
> > W5. Could you provide a detailed ablation separating the contributions of visual complexity evolution vs. textual complexity evolution?
>
> In **Table 9**, we presented results for **text-only evolution** (denoted as 'original data') and joint **text-image evolution** (denoted as 'complex data'). We additionally provide the results for image-only evolution (where questions remain unchanged) in the following table. Please note that all datasets in the table below consist of 0.6K samples. The 'visual-only evolution' setting corresponds to iteratively fine-tuning the model on the same set of data samples, but with the original images replaced by their more complex counterparts.
>
> | Model | Visual |
> | -------- | -------- |
> | C2-Evo-1st     | 46.2   |
> | C2-Evo-2nd     | 46.6   |
> | C2-Evo-3rd     | 46.4   |
>
>
> > W6. How does the computational cost of iterative evolution training compare to static dataset fine-tuning?
>
> We analyze the computational cost from two perspectives: model training and data generation.
>
> **Training Cost.** Our method is highly data-efficient. Compared to static fine-tuning methods like R1-VL which utilize approximately 120k samples, our approach uses a significantly smaller dataset (0.6k). This results in substantially lower computational overhead for model gradient updates.
>
> **Generation Cost.** While our pipeline consists of five stages (CoT generation, sub-problem generation, complex problem synthesis, description generation, and answer filtering), the token consumption for the latter four stages is notably lower than the initial CoT generation.
>
> To quantify this, let T denote the time required for CoT generation on 1k samples:
>
> **Static methods (e.g., R1-VL):** Generating CoT for 120k samples requires approximately **120T**.
>
> **Our method:** Even accounting for all five stages, one iteration consumes less than 4.8T. Over three iterations, the total consumption is less than **14.4T**.
>
> Furthermore, the time cost for code-to-image rendering is negligible. Consequently, the total computational cost of our iterative framework remains significantly lower than that of large-scale static fine-tuning baselines.
>
>  > W7. Missing reference: MMEvol: Empowering Multimodal Large Language Models with Evol-Instruct (ACL 2025)
>
> We have **added** this work to the "Self-improvement" subsection of the "Related Work" in the revised manuscript (**L140**).

---

### Author Response · Authors · 2025-11-24

Dear Area Chairs,

We thank all reviewers for their time and thoughtful feedback. We would like to express our sincere appreciation to the Area Chairs for your **extra works** on recommendation caused by the incident.
We appreciate the reviewers' recognition of C2-Evo, acknowledging it **fills an important gap in MLLM self-improvement research** (DRDG). The reviewers commended the method’s **generalization to broader multimodal benchmarks** (DRDG, j9to), noting its potential to be **applied to various fine-tuning scenarios** (bnAy). On the technical front, they highlighted that our paper **addresses the critical challenge of integrating the evolution of visual and textual data into a unified process** (KjBx), allowing for simultaneous evolution. Furthermore, the reviewers emphasized that our approach aligns with current research trends and achieves **significant performance gains with minimal data** (DRDG, j9to), while finding the manuscript to be **well-presented, clear, and well-organized** (KjBx, bnAy).


*Note: The brickred parts in the paper indicate content added in the revised version.*

## **Summary of feedback and rebuttal experiments:**

| Reviewer | Rating | Index | Corern |   Response |
| -------- | -------- | -------- | -------- | -------- |
| DRDG     | 6     | W1     | More experiments on other complex multimodal domains |  Point out the results on general metrics already presented in the paper (Table 2,7); Provide extra **experiments** to show significant gains in more practical scenarios (Table 15) ✔
|      |      | W2&Q1     | Dependence on GPT-4o |  Provide **findings** from different models during the experiments and **limitations of open-source models** in higher-order reasoning ✔
|      |      | W3     | Lack ablation on each guiding principle |  Adding one figure ✔
|      |      | Q2     | Lack ablation on visual vs. textual evolution | Point out the results on Visual+Textual vs. Textual evolution already presented in the paper (Table 9); Add results on Visual evolution (Table 17) ✔
|      |      | Q3     | Lack of comparison on computational cost | Provide an  **quantitative analysis** ✔
|      |      | Q4     | Missing reference | Adding the reference in L140 ✔
| j9to     | 4     | W1     | Imprecise description at L80 | Fixing the issue in updated paper ✔
|      |      | W2     | Questions about the reliability and cleanliness of generated data   |  Provide **experimental findings and effectiveness** on model selection and various filtering strategies ✔
|      |      | W3     | Limited task scope and scalability concerns | Point out the **results on general metrics already presented** in the paper (Table 2,7); Provide extra **experiments** to show significant gains in more practical scenarios (Table 15); Clarify that the framework’s optimization **objective is to increase data complexity rather than quantity**, and **the error-rate filter design** ensures efficient data utilization ✔
| KjBx     | 4     | W1&Q1     | Compared with papers MMEvol and MAmmoTh-VL | Point out that these **two papers fall under the first limitation**; Adding the reference in L140 ✔
|      |      | W2     |  Lack experiments on Qwen2.5-VL  | Add one figure （Table 16） ✔
|      |      | W3&Q2     | Lack experiments on general multimodal datasets; Question about the image diversity; Lack ablation on visual evolution | Point out the results on **general metrics already presented** in the paper (Table 2,7); Provide **findings on image diversity resulting from different methods** during the experiments; Point out results on **Visual+Textual vs. Textual evolution already presented** in the paper (Table 9) ✔
|      |      | Q3     | Dependence on GPT-4o; Compared with paper VILA^2 | Provide **findings** from different models during the experiments and **limitations of open-source models** in higher-order reasoning; Point out **the limitations of the VILA^2 paradigm in handling  higher-order reasoning tasks** and the superiority of C2-Evo ✔
| bnAy     | 4     | W1     | Incremental method incorporating SFT and GRPO  | Clarify our contribution **lies in the holistic design of the evolutionary closed-loop framework**, rather than proposing new SFT and GRPO algorithms ✔
|      |      | W2&Q1     |  Lack advanced baselines comparison  | Point out the **comparisons already presented in our paper** with three tables of various representative methods (Table 2,8,9) ✔


From the table above, we could find that:
* The primary concerns center on the need for **additional experiments, deeper analysis, clarification of technical design details, and more precise articulation**. We have provided all the requested information to comprehensively address these issues.


***It is unfortunate that a bug with OpenReview hindered our ability to fully engage in further discussion with the reviewers. We look forward to your assessment and final judgement.***

sincerely,

The authors

---

### Meta-Review · Area_Chair_Lwpd · 2026-01-10

**Summary:**

The main concerns from the reviewers are following:

- Reviewer **DRDG**:
  - **W1**: Evaluation primarily focuses on geometry problems.
  - **W2**: Heavy reliance on GPT-4o as an oracle for data generation.
  - **W3**: Limited ablation on individual components and visual/textual complexity evolution.

- Reviewer **j9to**:
  - **W1**: Noise can be introduced in the data evolution process.
  - **W2**: The work only addresses diagram-based mathematical reasoning tasks, with a relatively small dataset scale.

- Reviewer **KjBx**:
  - **W1**: Similar paradigm has been introduced in existing works.
  - **W2**: The base models in the experiments are outdated.
  - **W3**: The necessity of image evolution and the risk of introducing more errors.

- Reviewer **zkVT**:
  - **W1**: The proposed approach is somehow a combination of data-evolving and model-evolving methods.

**Reviewer Concerns:**

- **Concerns addressed in the rebuttal:**
  - The author responses address the issue of lacking ablations (**W3** of Reviewer **DRDG**), outdated base models (**W2** of Reviewer **KjBx**), and the difference to existing approaches (**W1** of Reviewer **zkVT**).

- **Concerns remained outstanding:**
  - I agree with **W2** of Reviewer **DRDG**: The data evolving process heavily relies on GPT-4o (or other strong enough proprietary models) , which is somehow engineering styled and lacks technical rigor. This also makes the risk of introducing noise and more errors (**W1** of Reviewer **j9to**, **W3** of Reviewer **KjBx**) difficult to address.

  - As pointed by **W1** of Reviewer **DRDG** and **W2** of Reviewer **j9to**, the experiments focuses on math reasoning. To justify that the proposed method is widely applicable to general multimodal reasoning, broader tasks and benchmarks are better to be included in the experiments.

**Reviewer Scores:**

As discussed in the Reviewer Concerns, some points are indeed addressed by the author responses. While in my view, the remaining concerns (**W1, W2** of Reviewer **DRDG**, **W1, W2** of Reviewer **j9to**, **W3** of Reviewer **KjBx**) also prevent the reviewers with negative evaluations to fully change their opinions.

---

### Decision · Program_Chairs · 2026-01-26

Reject